# Hyperbolic Active Learning for Semantic Segmentation under Domain Shift

## Abstract

We introduce a hyperbolic neural network approach to pixel-level active learning for semantic segmentation, and propose a novel geometric interpretation of the hyperbolic geometry that arises bottom-up from the statistics of the data. In our formulation the hyperbolic radius emerges as an estimator of the *unexplained class complexity*, which encompasses the class intrinsic complexity and its scarcity in the dataset. The unexplained class complexity serves as a metric indicating the likelihood that acquiring a particular pixel would contribute to enhancing the data information. We combine this quantity with prediction uncertainty to compute an acquisition score that identifies the most informative pixels for oracle annotation. Our proposed HALO (Hyperbolic Active Learning Optimization) sets a new state-of-the-art in active learning for semantic segmentation under domain shift, and surpasses the supervised domain adaptation performance while only using a small portion of labels (i.e., 1%). We perform extensive experimental analysis based on two established benchmarks, i.e. GTAV $\rightarrow$ Cityscapes and SYNTHIA $\rightarrow$ Cityscapes, and we additionally test on Cityscape $\rightarrow$ ACDC under adverse weather conditions.

## 1 Introduction

Dense prediction tasks, such as semantic segmentation (SS), are important in applications such as self-driving cars, manufacturing, and medicine. However, these tasks necessitate pixel-wise annotations, which can incur substantial costs and time inefficiencies (Cordts et al., 2016). Previous methods (Xie et al., 2022a; Vu et al., 2019; Shin et al., 2021b;a; Ning et al., 2021) have addressed this labeling challenge via domain adaptation, capitalizing on large source datasets for pre-training and domain-adapting with few target annotations (Ben-David et al., 2010). Most recently, active domain adaptation (ADA) has emerged as an effective strategy, i.e. annotating only a small set of target pixels in successive labelling rounds (Ning et al., 2021).

State-of-the-art (SoA) ADA relies on prediction uncertainty and pseudo-labels as the core strategy for active learning (AL) data acquisition (Shin et al., 2021b; Wu et al., 2022; Xie et al., 2022a). The current best performer (Xie et al., 2022a) introduces a region impurity score to prioritize the annotation of pixels likely at the class boundaries as a data acquisition strategy. But the pixels at the class boundaries are not necessarily the most informative and annotating only those degrades performance, as we confirm with an oracular study. Here, we argue that the scarcity of labels for certain class prototypical appearances and the intrinsic complexity of classes are better cues for an AL data acquisition strategy.

We propose Hyperbolic Active Learning Optimization (HALO), the first hyperbolic framework for AL, and a novel geometric interpretation of the hyperbolic radius. The SoA hyperbolic SS model (Atigh et al., 2022) trains with class hierarchies, which they manually define. As a result, their hyperbolic radius represents the parent-to-child hierarchical relations in the Poincaré ball. We adopt Atigh et al. (2022), but we find that hierarchies do not emerge naturally when they are not enforced at training time. E.g., in HALO *road* and *building* classes are closer to the center of the ball, while *person* and *rider* have larger radii. This class arrangement also defies the interpretation of the hyperbolic radius as a proxy for uncertainty, which emerged from metric learning hyperbolic studies (Ermolov et al., 2022; Franco et al., 2023), as *road* and *building* classes are not less uncertain. So neither interpretation explains the learned radii in the case of hierarchy-free hyperbolic SS.

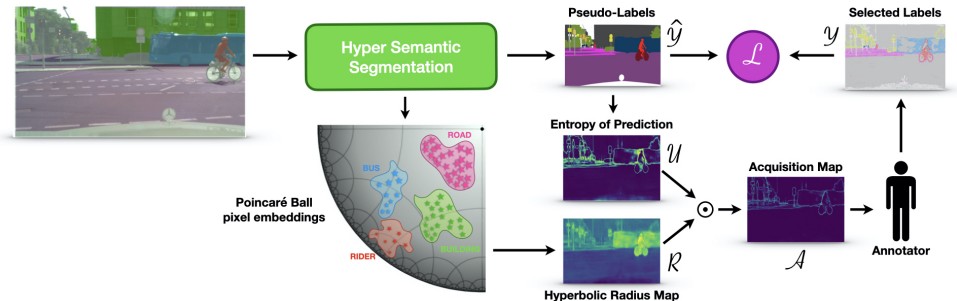

Figure 1: Overview of HALO. Pixels are encoded into the hyperbolic Poincaré ball and classified in the pseudo-label $\hat{y}$. The hyperbolic radius of the pixel embeddings defines the new hyperbolic score map $\mathcal{R}$. The uncertainty map $\mathcal{U}$ is extracted as the entropy of the classification probabilities. Combining $\mathcal{R}$ and $\mathcal{U}$ we define the data acquisition map $\mathcal{A}$, which is used to query new labels $\mathcal{Y}$.

We identify a novel interpretation of the hyperbolic geometry, wherein the hyperbolic radius serves as a proxy for the *unexplained class complexity*. This concept encompasses two facets: the intrinsic class complexity (for instance, a *rider* is more challenging to classify than the *road*), and the quantity of class labels the model has been exposed to during training (the *rider* class has fewer labels than *road*). Consider the HALO pipeline illustrated in Fig. 1 and the circular sector representing the Poincaré ball, where pixels from various classes are mapped. HALO learns a manifold where the distance of a class from the center is directly proportional to the unexplained class complexity. In Sec. 4 we show how the hyperbolic radius emerges bottom-up from data statistics as a proxy for the unexplained class complexity. Specifically, the radius correlates with the inherent complexity of the class and the scarcity of labeled data for it. In HALO, this motivates us to use the radius to directly acquire the most informative pixels during the active learning round.

We demonstrate the efficacy of our approach through extensive benchmarking on well-established datasets for SS via ADA as GTAV → Cityscapes, SYNTHIA → Cityscapes, and additionally testing on Cityscapes → ACDC under adverse weather conditions. HALO sets a new SoA on all the benchmarks and it surpasses the supervised domain adaptation baseline. Our framework also introduces a novel technique for enhancing the stability of hyperbolic training, which we refer to as *Hyperbolic Feature Reweighting* (HFR), cf. Sec. 5. Our code will be released.

In summary, our contributions include: 1) Presenting a novel geometric interpretation of the hyperbolic radius as a proxy for the concept of *unexplained class complexity*; 2) Introducing hyperbolic neural networks in AL and a novel pixel-based data acquisition score based on the hyperbolic radius; 3) Conducting a comprehensive analysis to validate both the concept and the algorithm while setting a new state-of-the-art across all the considered ADA benchmarks for SS.

## 2  RELATED WORKS

**Hyperbolic Representation Learning (HRL)**    Hyperbolic geometry has been extensively used to capture embeddings of tree-like structures (Nickel & Kiela, 2017; Chami et al., 2020) with low distortion Sala et al. (2018); Sarkar (2012). Since the seminal work of Ganea et al. (2018) on Hyperbolic Neural Networks (HNN), approaches have successfully combined hyperbolic geometry with model architectures ranging from convolutional (Shimizu et al., 2020) to attention-based (Gulcehre et al., 2018), including graph neural networks (Liu et al., 2019; Chami et al., 2019) and, most recently, vision transformers (Ermolov et al., 2022). There are two leading interpretations of the hyperbolic radius in hyperbolic space: as a measure of the prediction uncertainty (Chen et al., 2022; Ermolov et al., 2022; Franco et al., 2023) or as the hierarchical parent-to-child relation (Nickel & Kiela, 2017; Tifrea et al., 2018; Surís et al., 2021; Ermolov et al., 2022; Atigh et al., 2022). Our work builds on the SoA hyperbolic semantic segmentation method of Atigh et al. (2022), which enforces hierarchical labels and training objectives. However, when training hierarchy-free for ADA, as we do, the hierarchical interpretation does not apply; nor is the uncertainty viewpoint applicable. To the best of our knowledge, we are the first to propose a third interpretation for the HNNs connecting the hyperbolic space density to the semantic class recognition difficulty.

**Active Learning (AL)**    The number of annotations required for dense tasks such as semantic segmentation can be costly and time-consuming. Active learning balances the labeling efforts and performance, selecting the most informative pixels in successive learning rounds. Strategies for active learning are based on uncertainty sampling (Gal et al., 2017; Wang & Shang, 2014; Wang et al., 2016), diversity sampling (Ash et al., 2019; Kirsch et al., 2019; Sener & Savarese, 2017; Wu et al., 2021) or a combination of both (Sinha et al., 2019; Xie et al., 2022b; Prabhu et al., 2021; Xie et al., 2022a). For the case of AL in semantic segmentation, EqualAL (Golestaneh & Kitani, 2020) incorporates the self-supervisory signal of self-consistency to mitigate the overfitting of scenarios with limited labeled training data. Labor (Shin et al., 2021b) selects the most representative pixels within the generation of an inconsistency mask. PixelPick (Shin et al., 2021a) prioritizes the identification of specific pixels or regions over labeling the entire image. Mittal et al. (2023) explores the effect of data distribution, semi-supervised learning, and labeling budgets. We are the first to leverage the hyperbolic radius as a proxy for the most informative pixels to label next.

**Active Domain Adaptation (ADA)**    Domain Adaptation (DA) involves learning from a source data distribution and transferring that knowledge to a target dataset with a different distribution. Recent advancements in DA for semantic segmentation have utilized unsupervised (UDA) (Hoffman et al., 2018; Vu et al., 2019; Yang & Soatto, 2020; Liu et al., 2020; Mei et al., 2020; Liu et al., 2021) and semi-supervised (SSDA) (French et al., 2017; Saito et al., 2019; Singh, 2021; Jiang et al., 2020) learning techniques. However, challenges such as noise and label bias still pose limitations on the performance of DA methods. Active Domain Adaptation (ADA) aims to reduce the disparity between source and target domains by actively selecting informative data points from the target domain (Su et al., 2020; Fu et al., 2021; Singh et al., 2021; Shin et al., 2021b), which are subsequently labeled by human annotators. In semantic segmentation, Ning et al. (2021) propose a multi-anchor strategy to mitigate the distortion between the source and target distributions. The recent study of Xie et al. (2022a) shows the advantages of region-based selection in terms of region impurity and prediction uncertainty scores, compared to pixel-based approaches. By contrast, we show that selecting just from contours limits performance, and that *unexplained class complexity* is a better objective, as estimated by the hyperbolic radius.

## 3 BACKGROUND

We provide preliminaries on two techniques that HALO builds upon: Hyperbolic Image Segmentation  (Atigh et al., 2022) and Active Domain Adaptation.

**Hyperbolic Neural Networks**    We operate in the Poincaré ball hyperbolic space. We define it as the pair $(\mathbb{D}_c^N, g^{\mathbb{D}_c})$ where $\mathbb{D}_c^N = \{x \in \mathbb{R}^N : c\|x\| < 1\}$ is the manifold and $g_x^{\mathbb{D}_c} = (\lambda_x^c)^2 g^{\mathbb{E}}$ is the associated Riemannian metric, $-c$ is the curvature, $\lambda_x^c = \frac{2}{1-c\|x\|^2}$ is the conformal factor and $g^{\mathbb{E}} = \mathbb{I}^N$ is the Euclidean metric tensor. Hyperbolic neural networks first extract a feature vector $v$ in Euclidean space, which is subsequently projected into the Poincaré ball via exponential map:

$$exp_x^c(v) = x \oplus_c \left( \frac{v}{\sqrt{c}\|v\|} tanh \left( \sqrt{c}\frac{\lambda_x^c\|v\|}{2} \right) \right) \tag{1}$$

where $x \in \mathbb{D}_c^N$ is the anchor and $\oplus_c$ is the Möbius hyperbolic addition. The latter is defined for two hyperbolic vectors $h, w$ as follows:

$$h \oplus_c w = \frac{(1 + 2c\langle h, w \rangle + c\|w\|^2)v + (1 - c\|h\|^2)w}{1 + 2c\langle h, w \rangle + c^2\|h\|^2\|w\|^2} \tag{2}$$

We define the hyperbolic radius of the embedding $h \in \mathbb{D}_c^N$ as the Poincaré distance (See Eq. A1 in Appendix A.4) from the origin of the ball:

$$d(h, 0) = \frac{2}{\sqrt{c}} tanh^{-1} \left( \sqrt{c}\|h\| \right), \tag{3}$$

We propose to use the hyperbolic radius of the pixel embeddings as a novel data acquisition strategy. This is motivated by a novel geometric interpretation of the hyperbolic radius, which we support with experimental evidence in this section.

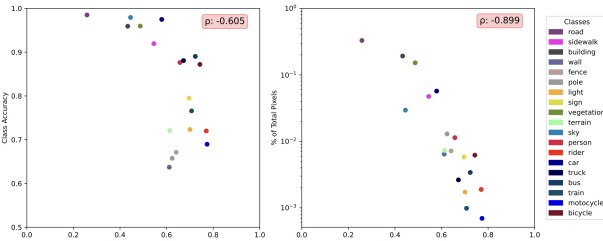

Figure 2: (left) Plot of average per-class radii of pixel embeddings Vs. class accuracies; (right) Plot of per-class radii of pixel embeddings Vs. the percentage of per-class target pixels.

**Hyperbolic Multinomial Logistic Regression (MLR)**   Following Ganea et al. (2018), to classify an image feature $z_i \in \mathbb{R}^N$ we project it onto the Poincaré ball $h_i = \exp_x^c(z_i) \in \mathbb{D}_c^N$ and classify with a number of hyperplanes $H_y^c$ (known as "gyroplanes") for each class $y$:

$$H_y^c = \{h_i \in \mathbb{D}_c^N, \langle -p_y \oplus_c h_i, w_y \rangle\}, \tag{4}$$

where, $p_y$ represents the gyroplane offset, and $w_y$ represents the orientation for class $y$. The distance between a Poincaré ball embedding $h_i$ and the gyroplane $H_y^c$ is given by:

$$d(h_i, H_y^c) = \frac{1}{\sqrt{c}} sinh^{-1} \left( \frac{2\sqrt{c}\langle -p_y \oplus_c h_i, w_y \rangle}{(1 - c\| - p_y \oplus_c h_i\|^2)\|w_y\|} \right), \tag{5}$$

Based on this distance, we define the likelihood as $p(\hat{y}_i = y | h_i) \propto exp(\zeta_y(h_i))$ where $\zeta_y(h_i) = \lambda_{p_y}^c \|w_y\| d(h_i, H_y^c)$ is the logit for the $y$ class.

**ADA for Semantic Segmentation**   The task aims to transfer knowledge from a source labeled dataset $\mathcal{S} = (X_s, Y_s)$ to a target unlabeled dataset $\mathcal{T} = (X_t, Y_t)$, where $X$ represents an image and $Y$ the corresponding annotation map. $Y_s$ is given, $Y_t$ is initially the empty set $\emptyset$. Adhering to the ADA protocol (Xie et al., 2022a; Wu et al., 2022; Shin et al., 2021b), target annotations are incrementally added in rounds, subject to a predefined budget, upon querying an annotator. Each pixel is assigned a priority score using a predefined acquisition map $\mathcal{A}$. Labels are added to $Y_s$ in each AL round by selecting pixels from $\mathcal{A}$ with higher scores, in accordance with the budget. The entire architecture undergoes end-to-end training, with back-propagation incorporating estimates $\hat{Y}_s$ and $\hat{Y}_t$ from the per-pixel cross-entropy loss $\mathcal{L}(\hat{Y}_s, \hat{Y}_t, Y_s, Y_t)$.

**Setup**   The work by Atigh et al. (2022) stands as the first to showcase hyperbolic semantic segmentation performance comparable to that of Euclidean networks. They proceed by mapping pixel embeddings onto a hyperbolic space, where they classify by hyperbolic multinomial logistic regression. We assume to have pre-trained the hyperbolic image segmenter of Atigh et al. (2022) on the source dataset GTAV (Richter et al., 2016) and to have domain-adapted it to the target dataset Cityscape (Cordts et al., 2016) with 5 rounds of AL, each adding 1% of the target labels. We assume to have followed the HALO pipeline of Fig. 1, which we detail in Sec. 5. The following section considers the radii of the hyperbolic pixel embeddings, for which statistics are computed on the Cityscape validation set.

## 4   HYPERBOLIC RADIUS AND THE UNEXPLAINED CLASS COMPLEXITY

In Sec. 4.1 we interpret the emerging properties of hyperbolic radius, and we compare with the interpretations in literature in Sec. 4.2.

### 4.1   EMERGING PROPERTIES OF THE HYPERBOLIC RADIUS

**What does the hyperbolic radius represent?**   Fig. 2 (left) illustrates the correlation between the per-class average hyperbolic radius and the relative class SS accuracy. They correlate negatively with a significant $\rho = -0.605$. So classes with larger hyperbolic radii have lower performance and are likely more difficult to recognize, more complex. E.g. *road* has large accuracy and small radius,

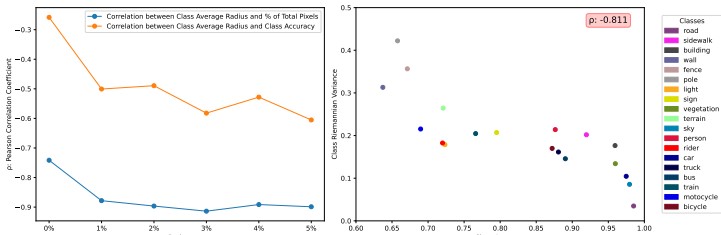

Figure 3: (left) Evolution of correlations between the class average radius and semantic segmentation accuracy (*orange*), and between the class average radius and the per-class percentage of total pixels (*blue*), at different budget levels during AL. (right) Plot of per-class accuracy against per-class Riemannian variance. Refer to Sec. 4 for detailed explanations.

*motocycle* has lower accuracy and larger radius. Fig. 2 (right) shows the correlation between the average class hyperbolic radius and the percentage of pixel labels for each class relative to the total number of pixels in the dataset. The correlation is substantial ($\rho = -0.899$), so classes with larger hyperbolic radii such as *motocycle* are rare in the target dataset, while at lower hyperbolic radii we have more frequent classes such as *road*. We conclude that *the hyperbolic radius indicates the difficulty in recognizing a class*, as a consequence of the class complexity and its label scarcity. Building upon this evidence, in Sec. 5, we introduce a novel acquisition score based on the hyperbolic radius to select pixels from classes that are inherently complex and rarer in the target dataset.

**How does learning the hyperbolic manifold of the pixels embeddings proceed?**     Fig. 3 (left) illustrates the evolution, during the active learning rounds, of the correlations between the per-class average radius and two quantities: the classification accuracy (orange), and the percentage of pixels belonging to the specified class in relation to the overall pixel count within the target dataset (blue). During training, both the correlations of the radius Vs. accuracy and the radius Vs. % of total pixels per class grow in module, confirming that the model progressively learns hyperbolic radii, indicative of the recognition difficulty of the class, based on the inherent complexity and label scarcity. The more HALO proceeds, the more the model is aware of what it does not know, i.e. HALO estimates what pixels it considers complex, which makes the best acquisition strategy.

**Novel geometric interpretation of the hyperbolic radius**     Fig. 3 (right) complements the findings by plotting the class accuracies Vs. the Riemannian variance (see Appendix A.4) of radii for each class. The latter generalizes the Euclidean variance, taking into consideration the increasing Poincaré ball density at larger radii. The correlation between accuracy and Riemannian variance is noteworthy ($\rho = -0.811$), indicating that challenging classes, like *pole*, exhibit lower accuracy and larger Riemannian variance, occupying a greater volume in the space. Our conclusion is that *the model achieves classification in the hyperbolic space by positioning complex classes at larger radii*, leveraging the denser space and increased volume to effectively model them.

## 4.2    COMPARING INTERPRETATIONS OF THE HYPERBOLIC RADIUS

It emerges from our analysis that larger radii are assigned to classes that are more difficult to recognize, for their inherent complexity and their label scarcity. Earlier work has explained the hyperbolic radius in terms of uncertainty or hierarchies. Techniques from the former (Chen et al., 2022; Ermolov et al., 2022; Franco et al., 2023) consider that the larger hyperbolic radii indicate more certain and unambiguous samples. This is typical of hyperbolic metric learning-based approaches, whereby the larger radius results in an exponentially larger matching penalty due to the employed Poincaré distance (See Eq. A1 in Appendix A.4). We argue that this yields a self-normalizing learning objective, effectively making the radius proportional to the errors, as those techniques show. Methods in favor of a hierarchical explanation (Nickel & Kiela, 2017; Tifrea et al., 2018; Surís et al., 2021; Ermolov et al., 2022; Atigh et al., 2022) consider hierarchical datasets, labeling, and classification objective functions. Hierarchies naturally align with the growing volume in the Poincaré ball, so children nodes from different parents are mapped further from each other than from their parents. Learning under hierarchical constraints results in leaf classes closer to the ball edge, and moving between them passes via their parents at lower hyperbolic radii. Our hyperbolic SS model is derived from Atigh et al. (2022) but it differs in the geometric meaning of the hyperbolic radii of pixel em-

Figure 4: (a) Original image; (b) Radius map depicting the hyperbolic radii of pixel embeddings; (c) Pixels (yellow) that have been selected for data acquisition. See Sec. 5 for details; (d) HALO prediction; (e) Ground Truth annotations. Zoom in for the details.

beddings. Our novel interpretation may emerge due to the use of the hyperbolic multinomial logistic regression objective without the enforced label hierarchies.

## 5 HYPERBOLIC ACTIVE LEARNING OPTIMIZATION (HALO)

This section outlines the HALO framework, which is founded on the novel interpretation of hyperbolic geometry. In Sec. 5.1 we review the HALO pipeline. In Sec. 5.2 we delve into the novel AL acquisition strategy based on the hyperbolic radius. In Sec. 5.3 we present our proposition for fixing the training instability of the hyperbolic framework.

### 5.1 HALO PIPELINE

Let us consider Fig. 1. During the training phase, we adhere to the hyperbolic semantic segmentation methodology presented by Atigh et al. (2022). However, we diverge from manually injecting hierarchies, as our approach relies exclusively on learning from data. The hyperbolic neural network used integrates an Euclidean Segmenter (e.g., DeepLabv3+), a hyperbolic projection layer (expmap), and a hyperbolic multinomial logistic regression (HyperMLR) layer. During the forward pass, the segmenter produces a $d$-dimensional embedding in Euclidean space for each pixel. Subsequently, each pixel embedding undergoes projection into the Poincaré ball via expmap. During the training phase, the HyperMLR is employed for classification based on the target labels selected in previous rounds of active learning.

At the conclusion of each training cycle, active learning is employed to identify the most informative pixels for annotation. Utilizing pixel embeddings, we estimate the hyperbolic radius $\mathcal{R}$ (as detailed in Sec. 4 and illustrated in Fig. 4b). Concurrently, predicted classification probabilities are used to compute pixel uncertainties $\mathcal{U}$, a technique inspired by prior works such as Paul et al. (2020); Shin et al. (2021a); Wang & Shang (2014); Wang et al. (2016); Xie et al. (2022a). New labels are then chosen based on a data acquisition score $\mathcal{A}$ (as depicted in Fig. 4c), calculated as the element-wise product of $\mathcal{R}$ and $\mathcal{U}$, and these labels are subsequently integrated into the training set. Note that the new labels are both at the boundaries and within, in areas with the largest inaccuracies (compare Fig. 4d and 4e). The rest of the ADA pipeline is as described in Sec. 3.

### 5.2 NOVEL DATA ACQUISITION STRATEGY

The acquisition score of each pixel in an image is formulated as the element-wise multiplication of the hyperbolic radii $\mathcal{R}$ and the uncertainties $\mathcal{U}$, i.e. $\mathcal{A} = \mathcal{R} \odot \mathcal{U}$. The radius $\mathcal{R}^{(i,j)}$ is computed as the distance of the hyperbolic pixel embedding $(i,j)$ from the center of the Poincaré ball (see Eq. 3):

$$\mathcal{R}^{(i,j)} = d(h_{i,j}, 0) = \frac{2}{\sqrt{c}} tanh^{-1} \left( \sqrt{c} \| h_{i,j} \| \right) \tag{6}$$

The uncertainty $\mathcal{U}^{(i,j)}$ is estimated as the entropy of the classification probability array $P_{i,j,c}$ associated with the pixel $(i,j)$ and the classes $c \in \{1, ..., C\}$:

$$\mathcal{U}^{(i,j)} = -\sum_{c=1}^{C} P_{i,j,c} \, log P_{i,j,c} \tag{7}$$

The acquisition score $\mathcal{A}$ serves as a surrogate indicator for the classification difficulty of each pixel and determines which pixels are presented to the human annotator for labeling, to augment the target label set $Y_t$.

Table 1: Comparison of mIoU results for different methods on the **GTAV → Cityscapes** task. Methods marked with ♯ are based on DeepLab-v3+ (Chen et al., 2018b), whereas all the others use DeepLab-v2 (Chen et al., 2018a).

| Method | road | side. | buil. | wall | fence | pole | light | sign | veg. | terr. | sky | pers. | rider | car | truck | bus | train | motor. | bike | mIoU |
|---|---|---|---|---|---|---|---|---|---|---|---|---|---|---|---|---|---|---|---|---|
| Eucl. Source Only | 75.8 | 16.8 | 77.2 | 12.5 | 21.0 | 25.5 | 30.1 | 20.1 | 81.3 | 24.6 | 70.3 | 53.8 | 26.4 | 49.9 | 17.2 | 25.9 | 6.5 | 25.3 | 36.0 | 36.6 |
| Hyper. Source Only | 62.4 | 18.7 | 66.8 | 17.4 | 13.8 | 29.2 | 30.4 | 7.4 | 83.2 | 23.8 | 78.2 | 56.1 | 30.3 | 70.6 | 25.0 | 17.8 | 0.3 | 27.6 | 27.0 | 36.1 |
| Hyper. Source Only♯ | 71.7 | 22.6 | 76.6 | 26.6 | 14.8 | 31.5 | 32.6 | 11.9 | 83.8 | 22.8 | 79.9 | 59.7 | 27.3 | 62.2 | 29.3 | 35.8 | 10.2 | 26.6 | 14.8 | 38.9 |
| CBST (Zou et al., 2018) | 91.8 | 53.5 | 80.5 | 32.7 | 21.0 | 34.0 | 28.9 | 20.4 | 83.9 | 34.2 | 80.9 | 53.1 | 24.0 | 82.7 | 30.3 | 35.9 | 16.0 | 25.9 | 42.8 | 45.9 |
| MRKLD (Zou et al., 2019) | 91.0 | 55.4 | 80.0 | 33.7 | 21.4 | 37.3 | 32.9 | 24.5 | 85.0 | 34.1 | 80.8 | 57.7 | 24.6 | 84.1 | 27.8 | 30.1 | 26.9 | 26.0 | 42.3 | 47.1 |
| Seg-Uncertainty (Zheng & Yang, 2021) | 90.4 | 31.2 | 85.1 | 36.9 | 25.6 | 37.5 | 48.8 | 48.5 | 85.3 | 34.8 | 81.1 | 64.4 | 36.8 | 86.3 | 34.9 | 52.2 | 1.7 | 29.0 | 44.6 | 50.3 |
| TPLD (Shin et al., 2020) | 94.2 | 60.5 | 82.8 | 36.6 | 16.6 | 39.3 | 29.0 | 25.5 | 85.6 | 44.9 | 84.4 | 60.6 | 27.4 | 84.1 | 37.0 | 47.0 | 31.2 | 36.1 | 50.3 | 51.2 |
| DPL-Dual (Cheng et al., 2021) | 92.8 | 54.4 | 86.2 | 41.6 | 32.7 | 36.4 | 49.0 | 34.0 | 85.8 | 41.3 | 86.0 | 63.2 | 34.2 | 87.2 | 39.3 | 44.5 | 18.7 | 42.6 | 43.1 | 53.3 |
| ProDA (Zhang et al., 2021) | 87.8 | 56.0 | 79.7 | 46.3 | 44.8 | 45.6 | 53.5 | 53.5 | 88.6 | 45.2 | 82.1 | 70.7 | 39.2 | 88.8 | 45.5 | 59.4 | 1.0 | 48.9 | 56.4 | 57.5 |
| WeakDA (point) (Paul et al., 2020) | 94.0 | 62.7 | 86.3 | 36.5 | 32.8 | 38.4 | 44.9 | 51.0 | 86.1 | 43.4 | 87.7 | 66.4 | 36.5 | 87.9 | 44.1 | 58.8 | 23.2 | 35.6 | 55.9 | 56.4 |
| LabOR (2.2%) (Shin et al., 2021b) | 96.6 | 77.0 | 89.6 | 47.8 | 50.7 | **48.0** | **56.6** | 63.5 | 89.5 | 57.8 | 91.6 | 72.0 | 47.3 | 91.7 | 62.1 | 61.9 | 48.9 | 47.9 | 65.3 | 66.6 |
| RIPU (2.2%) (Xie et al., 2022a) | 96.5 | 74.1 | 89.7 | 53.1 | 51.0 | 43.8 | 53.4 | 62.2 | 90.0 | 57.6 | 92.6 | 73.0 | 53.0 | **92.8** | 73.8 | **78.5** | 62.0 | **55.6** | 70.0 | 69.6 |
| **HALO (2.2%) (ours)** | **97.5** | **79.9** | **90.2** | **55.6** | **51.5** | 45.3 | 56.2 | **66.2** | **90.2** | **58.6** | **92.8** | **73.3** | **53.5** | 92.6 | **76.9** | 76.2 | **64.2** | 55.2 | **70.1** | **70.8** |
| AADA (5%)♯ (Su et al., 2020) | 92.2 | 59.9 | 87.3 | 36.4 | 45.7 | 46.1 | 50.6 | 59.5 | 88.3 | 44.0 | 90.2 | 69.7 | 38.2 | 90.0 | 55.3 | 45.1 | 32.0 | 32.6 | 62.9 | 59.3 |
| MADA (5%)♯ (Ning et al., 2021) | 95.1 | 69.8 | 88.5 | 43.3 | 48.7 | 45.7 | 53.3 | 59.2 | 89.1 | 46.7 | 91.5 | 73.9 | 50.1 | 91.2 | 60.6 | 56.9 | 48.4 | 51.6 | 68.7 | 64.9 |
| D²ADA (5%)♯ (Wu et al., 2022) | 97.0 | 77.8 | 90.0 | 46.0 | **55.0** | 52.7 | 58.7 | 65.8 | 90.4 | 58.9 | 92.1 | 75.7 | 54.4 | 92.3 | 69.0 | 78.0 | 68.5 | 59.1 | 72.3 | 71.3 |
| RIPU (5%)♯ (Xie et al., 2022a) | 97.0 | 77.3 | 90.4 | **54.6** | 53.2 | 47.7 | 55.9 | 64.1 | 90.2 | 59.2 | 93.2 | 75.0 | 54.8 | 92.7 | 73.0 | 79.7 | 68.9 | 55.5 | 70.3 | 71.2 |
| **HALO (5%)♯ (ours)** | **97.6** | **81.0** | **91.4** | 53.7 | 54.9 | **56.7** | **62.9** | **72.1** | **91.4** | **60.5** | **94.1** | **78.0** | **57.3** | **94.0** | **81.4** | **84.7** | **70.1** | **60.0** | **73.3** | **74.5** |
| Eucl. Supervised DA | 96.8 | 77.5 | 90.0 | 53.5 | 51.5 | 47.6 | 55.6 | 62.9 | 90.2 | 58.2 | 92.3 | 73.7 | 52.3 | 92.4 | 74.3 | 77.1 | 64.5 | 52.4 | 70.1 | 70.2 |
| Hyper. Supervised DA | 97.3 | 79.0 | 89.8 | 50.3 | 51.8 | 43.9 | 52.0 | 61.8 | 89.8 | 58.0 | 92.6 | 71.3 | 50.5 | 91.8 | 65.6 | 78.3 | 64.9 | 52.4 | 67.7 | 68.8 |
| Eucl. Supervised DA ♯ | 97.4 | 77.9 | 91.1 | 54.9 | 53.7 | 51.9 | 57.9 | 64.7 | 91.1 | 57.8 | 93.2 | 74.7 | 54.8 | 93.6 | 76.4 | 79.3 | 67.8 | 55.6 | 71.3 | 71.9 |
| Hyper. Supervised DA ♯ | 97.6 | 81.2 | 90.7 | 49.9 | 53.2 | 53.5 | 58.0 | 67.2 | 91.0 | 59.1 | 93.9 | 74.2 | 52.6 | 93.1 | 76.4 | 81.0 | 67.0 | 55.0 | 70.8 | 71.9 |

## 5.3 ROBUST HYPERBOLIC LEARNING WITH FEATURE REWEIGHTING

HNNs can be prone to stability issues during training because of the unique topology of the Poincaré ball. More precisely, when embeddings approach the boundary, the occurrence of vanishing gradients can impede the learning process. Several solutions have been proposed in the literature to address this problem (Guo et al., 2022; Franco et al., 2023; van Spengler et al., 2023). However, these approaches often yield sub-optimal or comparable performances when compared to the Euclidean counterpart. We introduce the *Hyperbolic Feature Reweighting (HFR)* module, designed to enhance training stability by reweighting features, prior to their projection onto the Poincaré ball. Given the feature map $Z \in \mathbb{R}^{\tilde{H} \times \tilde{W}}$ generated as the output from the encoder, we compute the weights as $L = \text{HRF}(Z) \in \mathbb{R}^{\tilde{H} \times \tilde{W}}$ and use them to rescale each entry of the normalized feature map, yielding $\tilde{Z} = \frac{Z}{|Z|} \odot L$, where $|Z| = \sum_{k=1}^{\tilde{H}\tilde{W}} z_{ij}$ and $\odot$ denotes the element-wise multiplication. Intuitively, reweighting increases the robustness as it prevents embeddings from getting too close to the boundaries, where the distances tend to infinity. Elsewhere, Guo et al. (2022) achieves robustness by clipping the largest values of the radii, Franco et al. (2023) makes it by curriculum learning, and van Spengler et al. (2023) needs to carefully initialize the hyperbolic network parameters. Our proposed HFR module is end-to-end trained and it enables the model to dynamically adapt through the various stages of training, endowing it with robustness.

## 6 RESULTS

In this section, we describe the benchmarks and training protocols; we perform a comparative evaluation against the SoA (Sec. 6.1); and we conduct ablation studies on the components, setups and hyper-parameters of HALO (Sec. 6.2). The implementation follows Xie et al. (2022a) and it is detailed in Appendix A.5.

**Datasets** The model has been pre-trained using synthetic cityscapes images from the GTAV (Richter et al., 2016) and SYNTHIA (Ros et al., 2016) datasets. The **GTAV** dataset contains 24,966 high-resolution frames that are densely labeled and divided into 19 classes that are fully compatible with the Cityscapes dataset. The **SYNTHIA** dataset includes a selection of 9,000 images with a resolution of $1280 \times 760$ and 16 classes. For ADA training and evaluation we consider the real-world urban street scenes from **Cityscapes** or **ACDC** as target datasets, both categorized into the same 19 classes. The **Cityscapes** (Cordts et al., 2016) dataset consists of 2,975 training samples and 500 validation samples. These images are of high resolution, with dimensions of $2048 \times 1024$. The **ACDC** (Sakaridis et al., 2021) dataset comprises 4,006 images captured under adverse conditions (i.e., fog, nighttime, rain, snow) to maximize the complexity and diversity of the scenes.

Table 2: Comparison of mIoU results for different methods on the **SYNTHIA → Cityscapes** task. Methods marked with ♯ are based on DeepLab-v3+ (Chen et al., 2018b), whereas all the others use DeepLab-v2 (Chen et al., 2018a).

| Method | road | side. | buil. | wall* | fence* | pole* | light | sign | veg. | sky | pers. | rider | car | bus | motor. | bike | mIoU | mIoU* |
|---|---|---|---|---|---|---|---|---|---|---|---|---|---|---|---|---|---|---|
| Eucl. Source Only | 64.3 | 21.3 | 73.1 | 2.4 | 1.1 | 31.4 | 7.0 | 27.7 | 63.1 | 67.6 | 42.2 | 19.9 | 73.1 | 15.3 | 10.5 | 38.9 | 34.9 | 40.3 |
| Hyper. Source Only | 36.4 | 21.1 | 56.4 | 13.3 | 0.1 | 24.8 | 0.0 | 9.5 | 78.8 | 70.4 | 54.2 | 8.6 | 77.9 | 35.8 | 11.7 | 27.3 | 32.9 | 37.5 |
| Hyper. Source Only♯ | 60.5 | 27.4 | 75.2 | 13.3 | 0.3 | 31.4 | 0.0 | 23.2 | 79.3 | 68.1 | 57.8 | 18.7 | 61.3 | 27.3 | 10.3 | 23.5 | 36.1 | 41.0 |
| CBST (Zou et al., 2018) | 68.0 | 29.9 | 76.3 | 10.8 | 1.4 | 33.9 | 22.8 | 29.5 | 77.6 | 78.3 | 60.6 | 28.3 | 81.6 | 23.5 | 18.8 | 39.8 | 42.6 | 48.9 |
| MRKLD (Zou et al., 2019) | 67.7 | 32.2 | 73.9 | 10.7 | 1.6 | 37.4 | 22.2 | 31.2 | 80.8 | 80.5 | 60.8 | 29.1 | 82.8 | 25.0 | 19.4 | 45.3 | 43.8 | 50.1 |
| DPL-Dual (Cheng et al., 2021) | 87.5 | 45.7 | 82.8 | 13.3 | 0.6 | 33.2 | 22.0 | 20.1 | 83.1 | 86.0 | 56.6 | 21.9 | 83.1 | 40.3 | 29.8 | 45.7 | 47.0 | 54.2 |
| TPLD (Shin et al., 2020) | 80.9 | 44.3 | 82.2 | 19.9 | 0.3 | 40.6 | 20.5 | 30.1 | 77.2 | 80.9 | 60.6 | 25.5 | 84.8 | 41.1 | 24.7 | 43.7 | 47.3 | 53.5 |
| Seg-Uncertainty (Zheng & Yang, 2021) | 87.6 | 41.9 | 83.1 | 14.7 | 1.7 | 36.2 | 31.3 | 19.9 | 81.6 | 80.6 | 63.0 | 21.8 | 86.2 | 40.7 | 23.6 | 53.1 | 47.9 | 54.9 |
| ProDA (Zhang et al., 2021) | 87.8 | 45.7 | 84.6 | 37.1 | 0.6 | 44.0 | 54.6 | 37.0 | 88.1 | 84.4 | 74.2 | 24.3 | 88.2 | 51.1 | 40.5 | 45.6 | 55.5 | 62.0 |
| WeakDA (point) (Paul et al., 2020) | 94.9 | 63.2 | 85.0 | 27.3 | 24.2 | 34.9 | 37.3 | 50.8 | 84.4 | 88.2 | 60.6 | 36.3 | 86.4 | 43.2 | 36.5 | 61.3 | 57.2 | 63.7 |
| RIPU (2.2%) (Xie et al., 2022a) | 96.8 | 76.6 | 89.6 | 45.0 | 47.7 | 45.0 | 53.0 | 62.5 | 90.6 | **92.7** | 73.0 | 52.9 | 93.1 | 80.5 | 52.4 | 70.1 | 70.1 | 75.7 |
| **HALO (2.2%) (ours)** | **97.5** | **81.7** | **90.5** | **52.8** | **52.8** | **45.6** | **57.3** | **67.1** | **91.2** | 92.6 | **74.5** | **54.9** | **93.3** | **81.6** | **55.2** | **71.1** | **72.5** | **77.6** |
| AADA (5%)♯ (Su et al., 2020) | 91.3 | 57.6 | 86.9 | 37.6 | 48.3 | 45.0 | 50.4 | 58.5 | 88.2 | 90.3 | 69.4 | 37.9 | 89.9 | 44.5 | 32.8 | 62.5 | 61.9 | 66.2 |
| MADA (5%)♯ (Ning et al., 2021) | 96.5 | 74.6 | 88.8 | 45.9 | 43.8 | 46.7 | 52.4 | 60.5 | 89.7 | 92.2 | 74.1 | 51.2 | 90.9 | 60.3 | 52.4 | 69.4 | 68.1 | 73.3 |
| D²ADA (5%)♯ (Wu et al., 2022) | 96.7 | 76.8 | 90.3 | 48.7 | 51.1 | 54.2 | 58.3 | 68.0 | 90.4 | 93.4 | 77.4 | 56.4 | 92.5 | 77.5 | 58.9 | 73.3 | 72.7 | 77.7 |
| RIPU (5%)♯ (Xie et al., 2022a) | 97.0 | 78.9 | 89.9 | 47.2 | 50.7 | 48.5 | 55.2 | 63.9 | 91.1 | 93.0 | 74.4 | 54.1 | 92.9 | 79.9 | 55.3 | 71.0 | 71.4 | 76.7 |
| **HALO (5%)♯ (ours)** | **97.5** | **81.5** | **91.5** | **56.5** | **52.7** | **57.0** | **63.2** | **72.9** | **92.0** | **94.4** | **77.8** | **57.4** | **94.4** | **86.1** | **60.5** | **73.5** | **75.6** | **80.2** |
| Eucl. Supervised DA | 96.7 | 77.8 | 90.2 | 40.1 | 49.8 | 52.2 | 58.5 | 67.6 | 91.7 | 93.8 | 74.9 | 52.0 | 92.6 | 70.5 | 50.6 | 70.2 | 70.6 | 75.9 |
| Hyper. Supervised DA | 97.6 | 81.9 | 90.2 | 52.0 | 49.6 | 45.5 | 51.7 | 65.0 | 90.9 | 93.0 | 73.1 | 50.3 | 92.6 | 80.7 | 50.8 | 69.2 | 70.9 | 75.9 |
| Eucl. Supervised DA♯ | 97.5 | 81.4 | 90.9 | 48.5 | 51.3 | 53.6 | 59.4 | 68.1 | 91.7 | 93.4 | 75.6 | 51.9 | 93.2 | 75.6 | 52.0 | 71.2 | 72.2 | 77.1 |
| Hyper. Supervised DA♯ | 97.7 | 82.2 | 90.3 | 53.0 | 48.8 | 51.7 | 56.0 | 66.1 | 91.4 | 94.2 | 75.0 | 51.5 | 93.4 | 82.1 | 52.8 | 70.2 | 72.3 | 77.1 |

Table 3: Comparison of mIoU results for HALO (ours) and RIPU (Xie et al., 2022a) on the **Cityscapes → ACDC** task. Methods marked with ♯ are based on DeepLab-v3+ (Chen et al., 2018b), whereas all the others use DeepLab-v2 (Chen et al., 2018a).

| Method | road | side. | buil. | wall | fence | pole | light | sign | veg. | terr. | sky | pers. | rider | car | truck | bus | train | motor. | bike | mIoU |
|---|---|---|---|---|---|---|---|---|---|---|---|---|---|---|---|---|---|---|---|---|
| RIPU (2.2%) | 91.4 | 69.5 | 83.8 | **52.7** | 41.6 | 52.8 | 66.4 | 54.2 | **85.1** | 47.5 | **94.7** | 54.5 | 21.8 | **85.5** | **58.7** | 58.8 | **76.9** | 41.4 | **45.9** | 62.3 |
| **HALO (2.2%)** | **92.6** | **71.3** | **84.5** | 51.3 | **43.1** | **53.5** | **67.2** | **57.6** | 85.1 | **49.5** | 94.5 | **57.2** | **28.6** | 84.1 | 53.3 | 76.0 | 66.9 | **44.1** | 41.4 | **63.2** |
| RIPU (5%)♯ | **92.7** | **72.5** | 84.7 | 53.1 | 44.8 | 56.7 | 69.1 | 58.9 | 85.9 | 46.9 | **95.3** | 57.2 | 24.3 | 84.5 | **61.4** | 59.4 | 79.0 | 36.9 | 43.6 | 63.5 |
| **HALO (5%)♯** | 92.6 | 72.2 | **84.8** | **54.9** | **47.7** | **59.5** | **71.5** | **61.1** | **86.1** | 49.5 | 95.2 | **60.7** | **30.6** | **85.8** | 58.4 | **73.8** | **82.0** | **41.6** | **53.2** | **66.4** |

**Training protocols** The models undergo a source-only pre-training on either GTAV or SYN-THIA synthetic datasets. To compare and evaluate the performance with other methods, two ADA protocols are used: source-free and source+target. In the source-free protocol, only the Cityscapes dataset is used, whereas in the source+target protocol, both source and target datasets are utilized. In both protocols, our hyperbolic radius-based selection method is used to select pixels to be labeled in five evenly spaced rounds during training, with either 2.2% or 5% of total pixels selected. Supervised DA models are trained for comparison purposes with active learning protocols. Our model is additionally trained under adverse conditions, using Cityscapes and ACDC as the source and target datasets respectively, in line with Hoyer et al. (2023) and Brüggemann et al. (2023).

**Evaluation metrics** To assess the effectiveness of the models, the mean Intersection-over-Union (mIoU) metric is computed on the target validation set. For GTAV-Cityscapes and Cityscapes-ACDC, the mIoU is calculated on the shared 19 classes, whereas for SYNTHIA-Cityscapes two mIoU values are reported, one on the 13 common classes (mIoU*) and another on the 16 common classes (mIoU).

## 6.1 COMPARISON WITH THE STATE-OF-THE-ART

In Table 1, we present the results of our method and the most recent ADA approaches on the GTAV → Cityscapes benchmark with the source+target protocol. HALO outperforms the current state-of-the-art methods (RIPU Xie et al. (2022a), D²ADA Wu et al. (2022)) using both 2.2% (+1.2% mIoU) and 5% (+3.3% mIoU) of labeled pixels, reaching 70.8% and 74.5%, respectively. Additionally, our method is the first to surpass the supervised domain adaptation baseline (71.9%), even by a significant margin (+2.6%). HALO achieves state-of-the-art also in the SYNTHIA → Cityscapes case (cf. Table 2), where it improves by +2.4% and +4.2% using 2.2% and 5% of labels, reaching performances of 72.5% and 75.6%, respectively. HALO also surpasses the current best Xie et al. (2022a) by +3% in the source-free scenario, achieving performances close to the source+target with 5% budget (73.3% vs. 74.5%), as shown in Table 4. Due to the absence of other ADA studies on the Cityscapes to ACDC adaptation, we trained RIPU (Xie et al., 2022a) as a baseline for comparison

Table 4: HALO performance on the **source-free protocol** compared with previous UDA and ADA approaches.

| Method | Budget | mIoU |
|---|---|---|
| URMA (S & Fleuret, 2021) | - | 45.1 |
| LD (You et al., 2021) | - | 45.5 |
| SFDA (Kundu et al., 2021) | - | 53.4 |
| RIPU (Xie et al., 2022a) | 2.2% | 67.1 |
| **HALO** (ours) | 2.2% | **70.1** |
| **HALO**$^\sharp$ (ours) | 5% | **73.3** |

Table 5: Ablation study conducted with the Hyperbolic DeepLab-v3+ as backbone on the source+target protocol with 5% budget. Performance of entropy and hyperbolic radius scores in isolation (a and b) and combined (c).

| Ablative version | mIoU |
|---|---|
| (a) Entropy only | 63.2 |
| (b) Hyperbolic Radius only | 64.1 |
| (c) **Hyperbolic Radius $\odot$ Entropy (HALO)** | **74.5** |

with our method. HALO demonstrates superiority over RIPU by +0.9% in the source+target setup with a 2.2% budget, and by +2.9% with a 5% budget, reaffirming the effectiveness of our approach on a novel dataset, as shown in Table 3. Certain classes may show unstable performance, attributed to the dataset's difficulty, requiring specialized methods (Brüggemann et al., 2023).

## 6.2 Ablation Study

We conduct ablation studies on the selection criteria, region- and pixel-based acquisition scores, labeling budget, reported next, and on the HFR, in the Appendix.

**Selection criteria** HALO demonstrates a substantial improvement of +10.4% compared to methods (a) and (b) in Table 5. More precisely, utilizing solely either the entropy (a) or the hyperbolic radius (b) as acquisition scores yields comparable performance of 63.2% and 64.1%, respectively. When these two metrics are combined, the final performance is notably improved to 74.5%.

**Region- Vs. Pixel-based criteria** Unlike region impurity in Xie et al. (2022a), the hyperbolic radius is a continuous quantity that can be computed for each pixel. We conduct experiments comparing region- and pixel-based acquisition scores. The results demonstrate a small difference between the two approaches (74.1% Vs. 74.5%).

**Labeling budget** We experiment with different labeling budgets, observing performance improvements as the number of labeled pixels increases. However, beyond a threshold of 5%, adding more labeled pixels leads to diminishing returns. We believe this may be explained by data unbalance: taking all labels to domain adapt means that most of them belong to a few classes, specifically *road*, *building* and *vegetation* account for 77% of the labels, which may hinder at successive training rounds due to data redundancy. Detailed results are in Fig. 5.

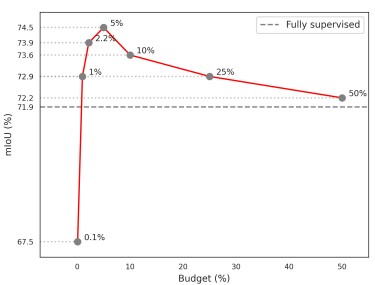

Figure 5: Performance on GTAV $\rightarrow$ Cityscapes with different budgets.

**Hyperbolic Feature Reweighting (HFR)** HFR improves training stability and enhances performance in the Hyperbolic model. Although the mIoU improvement is modest (+2%), the main advantage is the training robustness, as the Hyperbolic model otherwise struggles to converge. HFR does not benefit the Euclidean model and instead negatively impacts its performance. Additional results in Appendix A.1

## 7 Conclusions

We have introduced the first hyperbolic neural network technique for active learning, which we have extensively validated as the novel SoA on semantic segmentation under domain shift. We have identified a novel geometric interpretation of the hyperbolic radius, distinct from the established hyperbolic uncertainty and hyperbolic hierarchy, and we have supported the finding with experimental evidence. The novel concept of hyperbolic radius and its successful use as data acquisition strategy in AL are a step forward in understanding hyperbolic neural networks.

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

APPENDIX

This appendix provides additional information and insights on the proposed Hyperbolic Active Learning Optimization (HALO) for semantic segmentation under domain shift.

This supplementary material is structured as follows:

**A.1: Additional Ablation Studies** presents additional ablation studies on the embedding dimensions and the proposed Hyperbolic Feature Reweighting (HFR) for Euclidean and hyperbolic backbones;

**A.2 Limitations** discusses current limitations and recommendations for further research;

**A.3 Broader Impact** regards future applications and the fostered progress in the field of Hyperbolic Neural Networks;

**A.4 Additional Hyperbolic Formulas** reports additional employed hyperbolic formulas;

**A.5 Implementation Details** describes the training details adopted in the experiments;

**A.6 Qualitative Results** showcases representative qualitative results of HALO;

**A.7 Data Acquisition Strategy: rounds of selections** illustrates examples of pixel labeling selection and the priorities of the data acquisition strategy at each acquisition round;

**A.8 Additional Comparison with baseline model** illustrates a qualitative comparison of pixel acquisition between HALO and baseline model and oracle experiment to prove the limitation of boundary-only selection;

**A.9 Additional Correlation Analyses** presents three analyses on correlations on Cityscapes $\rightarrow$ ACDC benchmark, data unbalancing when budget increases and the correlations with the uncertainty score.

## A.1 ADDITIONAL ABLATION STUDIES

**Results of HFR**    Table A1 provides insights into the performance of hyperbolic and Euclidean models with and without Hyperbolic Feature Reweighting (HFR). In the case of HALO, the performance with and without HFR remains the same in the source-only setting. However, when applied to the source+target ADA scenario, HFR leads to an improvement of 1.2%. It should be noted that HFR also stabilizes the training of hyperbolic models. In fact, when not using HFR, training requires a warm-up schedule and, still, it does not converge in approximately 20% of the runs. HFR improves therefore performance for ADA and it is important for hyperbolic learning stability.

Table A1: **HFR Performance Comparison:** Evaluating the impact of Hyperbolic Feature Reweighting (HFR) on hyperbolic and Euclidean models in source-only and source+target protocols.

| Encoder | Protocol | HFR | mIoU (%) |
|---|---|---|---|
| DeepLab-v3+ | source-only | ✗ | 36.3 |
| DeepLab-v3+ | source-only | ✓ | 22.7 |
| Hyper DeepLab-v3+ | source-only | ✗ | 39.0 |
| Hyper DeepLab-v3+ | source-only | ✓ | 38.9 |
| HALO | source+target | ✗ | 72.5 |
| **HALO** | **source+target** | ✓ | **74.5** |

## A.2 LIMITATIONS

While we have presented experimental evidence supporting the need for a novel interpretation of the hyperbolic radius, our work lacks a rigorous mathematical validation of the properties of the hyperbolic radius within the given experimental setup. Future research should delve into this mathematical aspect to formalize and prove these properties.

HALO's reliance on a source model pretrained on synthetic data introduces challenges related to large-scale simulation efforts and the need for effective synthetic-to-real domain adaptation. Exploring alternative strategies, such as self-supervised pre-training on real source datasets, could be a promising research direction to mitigate these challenges.

Although Active Domain Adaptation significantly reduces labeling costs, the manual annotation of individual pixels can be a time-consuming task. Further investigation into human-robot interaction methodologies to streamline pixel annotation processes and expedite the annotation workflow is needed.

### A.3 BROADER IMPACT

Hyperbolic Neural Networks (HNN) have recently become mainstream, reaching state-of-the-art across several tasks. Still, the theory and interpretation of HNN is diverse across tasks. Specifically, the hyperbolic radius has been interpreted as a continuum hierarchical parent-to-child measure or as an estimate of uncertainty. Our novel third way of interpreting the radius adds to the flourishing framework of HNN, making a step forward.

### A.4 ADDITIONAL HYPERBOLIC FORMULAS

Here we report established hyperbolic formulas which have used in the paper, but not shown due to space constraints.

**Poincaré Distance** Given two hyperbolic vectors x, y $\in \mathbb{D}_c^N$, the *Poincaré distance* represents the distance between them in the Poincaré ball and is defined as:

$$d_{Poin}(x, y) = \frac{2}{\sqrt{c}} tanh^{-1}(\sqrt{c}\| - x \oplus_c y\|) \tag{A1}$$

where $\oplus_c$ is the Möbius addition defined in Eq. 2 of the paper and $c$ is the manifold curvature.

**Riemannian Variance** Given a set of hyperbolic vectors $x_1, ..., x_M \in \mathbb{D}_c^N$ we define the Riemannian variance between them as:

$$\sigma^2 = \frac{1}{M} \sum_{i=1}^{M} d_{Poin}^2(x_i, \mu) \tag{A2}$$

where $\mu$ is the Fréchet mean, the hyperbolic vector that minimizes the Riemannian variance. $\mu$ cannot be computed in closed form, but it may be approximated with a recursive algorithm Lou et al. (2021).

### A.5 IMPLEMENTATION DETAILS

For all experiments, the model is trained on 4 Tesla V100 GPUs using PyTorch Paszke et al. (2019) and PyTorch Lightning with an effective batch-size of 8 samples (2 per GPU). The DeepLab-v3+ architecture is used with an Imagenet pre-trained ResNet-101 as the backbone. *RiemannianSGD* optimizer with momentum of 0.9 and weight decay of $5 \times 10^{-4}$ is used for all the trainings. The base learning rates for the encoder and decode head are $1 \times 10^{-3}$ and $1 \times 10^{-2}$ respectively, and they are decayed with a "polynomial" schedule with power 0.5. The models are pre-trained for 15K iterations and adapted for an additional 15K on the target set. As per Xie et al. (2022a), the source images are resized to $1280 \times 720$, while the target images are resized to $1280 \times 640$.

### A.6 QUALITATIVE RESULTS

In Fig. 8, we present visualizations of HALO's predicted segmentation maps and the selected pixels. In the first row, HALO prioritizes the selection of pixels that are not easily interpretable, as evident in the *fence* or *wall* on the right side of the image. Notably, HALO does not limit itself to selecting contours exclusively; it continues to acquire pixels within classes if they exhibit high *unexplained class complexity* (determined by the hyperbolic radius). This behavior is also observed in rows 2,

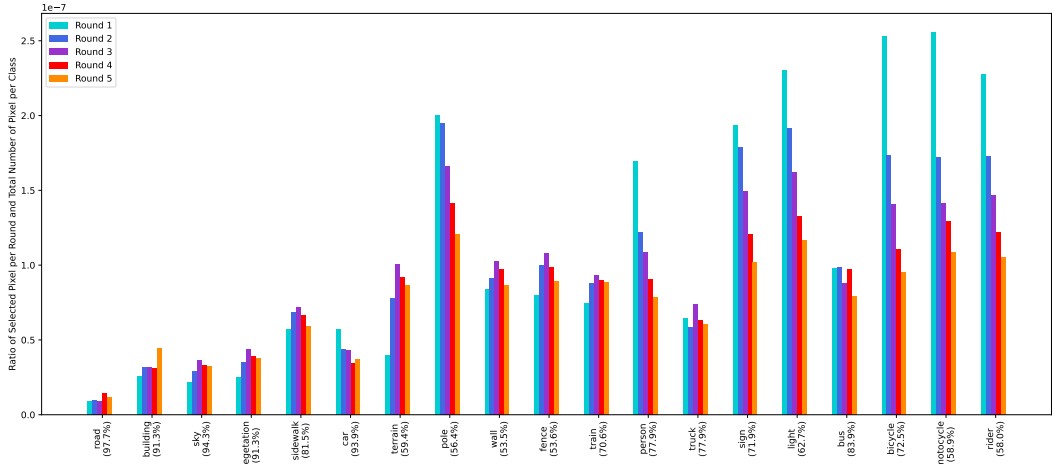

Figure 6: Ratio between the selected pixels for each class at each round and the total number of pixels per class. Each color shows the ratio in the specific round. On the $x$-axis are reported the classes with the relative mIoU (%) of HALO (cf. Table 1 of the main paper) ordered according to they decreasing hyperbolic radius.

3, and 4 of Fig. 8. For classes with lower complexity, such as *road* and *car*, HALO acquires only the contours. However, for more intricate classes like *pole* and *signs*, it also selects pixels within the class.

In rows 5, 6, and 8, the images depict a crowded scene with numerous small objects from various classes. Remarkably, the selection process directly targets the more complex classes (such as *pole* and *signs*), providing an accurate classification of these. In row 7, we observe an example where the most common classes (*road*, *vegetation*, *building*, *sky*) dominate the majority of the image. HALO, guided by the concept of *unexplained class complexity*, efficiently allocates the label budget by focusing on the more complex classes, rather than expending resources on these prevalent ones. Refer to Sec. A.7 and Fig. 7 for a detailed overview of the selection prioritization during each active learning round.

## A.7   DATA ACQUISITION STRATEGY: ROUNDS OF SELECTIONS

In this section, we analyze how the model prioritizes the selection of the pixels during the different rounds. In Fig. 6, we consider the ratio between the selected pixel at each round and the total number of pixels for the considered class. Note how the model selects in the early stages from the class with high intrinsic difficulty (e.g., rider, bicycle, pole). During the different rounds, the selected pixels decrease because of the scarcity of pixels associated with these classes. On the other hand, the classes with lower *unexplained class complexity* are less considered in the early stages and the model selects from them in the intermediate rounds if the class has an intermediate complexity (e.g., wall, fence, sidewalk) or in the last stages if the classes have low complexity (e.g., road or building).

The qualitative samples of pixel selections in Fig. 7 corroborate this observation. In rounds 1 and 2, the model gives precedence to selecting pixels from classes exhibiting high "unexplained class complexity" (e.g., *poles*, *sign*, *person*, or *rider*). Subsequently, HALO shifts its focus to two distinct objectives: i) acquiring contours from classes with lower complexity (e.g., *road*, *car*, or *vegetation*), and ii) obtaining additional pixels from more complex classes (e.g., *pole* or *wall*). Notably, in rows 1, 2, 3, 5, and 6, HALO gives priority to selecting complete objects right from the initial round (as seen with the *sign*). Another noteworthy instance is the acquisition of the *bicycle* in row 7. The hyperbolic radius score enables the acquisition of contours that extend beyond the boundaries of pseudo-label classes. In this case, we observe precise delineation of the internal portions of the wheels.

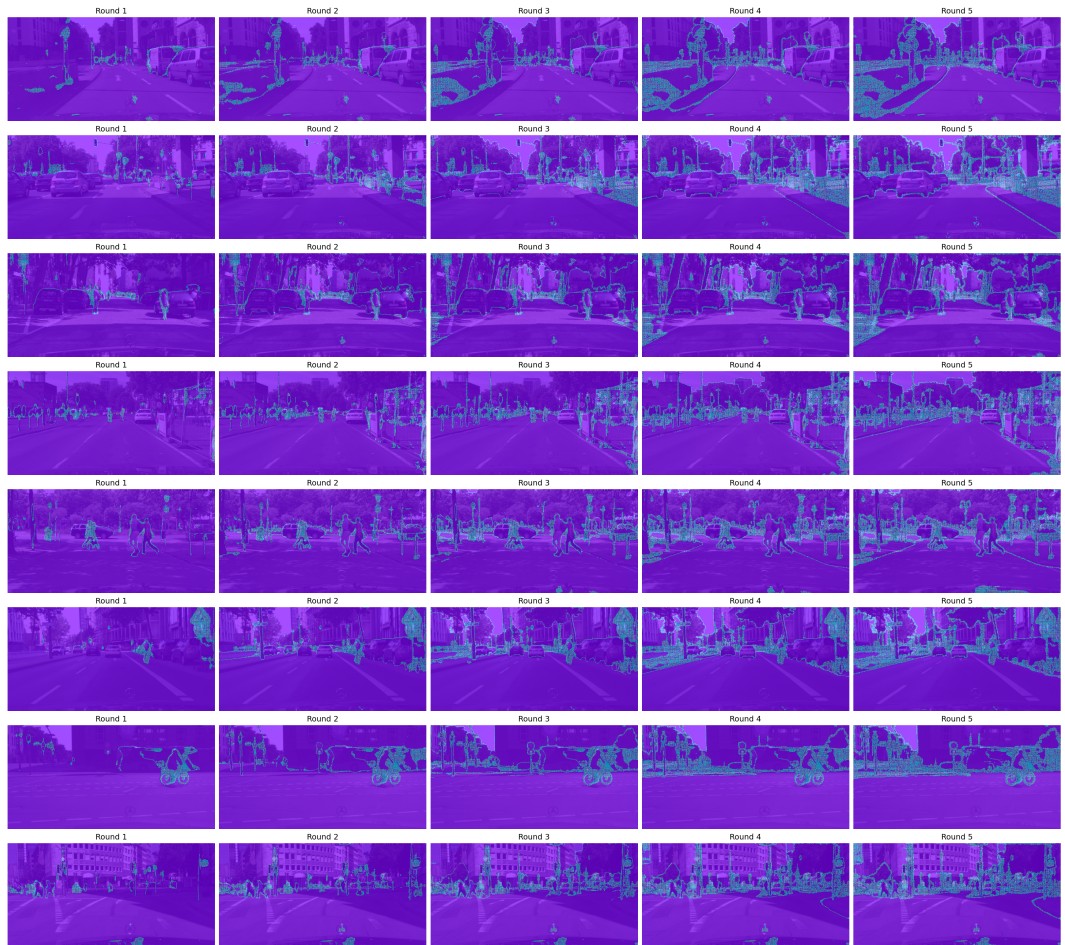

Figure 7: Qualitative analysis on the pixel selected by HALO at each round. Zoom in to see details.

## A.8 ADDITIONAL COMPARISON WITH THE BASELINE MODEL

**Qualitative Selection Comparison**    The top row of Fig. 9 depicts label acquisition using the baseline RIPU method with budgets of 2.2% (left) and 5% (right). The bottom row illustrates visualizations with our proposed HALO using the same budgets. Noteworthy observations include:

- By design, RIPU only concentrates on selecting boundaries between semantic parts (ref. Fig. 9 top-left). However, since there are only a few (thin) boundary pixels, RIPU soon exhausts the pixel selection request. Next, when a larger budget is available, RIPU simply samples from the left side. The random selection still provides additional labels (ref. Fig. 9 top-right) and is a good baseline, cf. Table 3 of Xie et al. (2022a), although not as good as HALO's acquisition strategy.
- By contrast, HALO showcases pixel selection from both boundaries and internal regions within semantic parts (ref. Fig. 9 bottom-left). Especially passing from 2.2% to 5% acquisition budget, HALO considers thick boundaries, so also parts of objects close to the boundaries, but also areas within objects, as it happens for wall, fence, pole, and sidewalk, cf. the right image part in the bottom-right of Fig. 9.

**Oracle Experiment with ground-truth boundaries**    We replace in RIPU (Xie et al., 2022a) the pseudo-labels with ground-truth ones, so we test an active learning acquisition strategy solely based on ground-truth boundary pixels. Although oracular, the experiment yields a performance drop of 1.4 mIoU (69.8 mIoU compared to RIPU's 71.2 mIoU), which validates HALO's label selection approach from non-boundary regions.

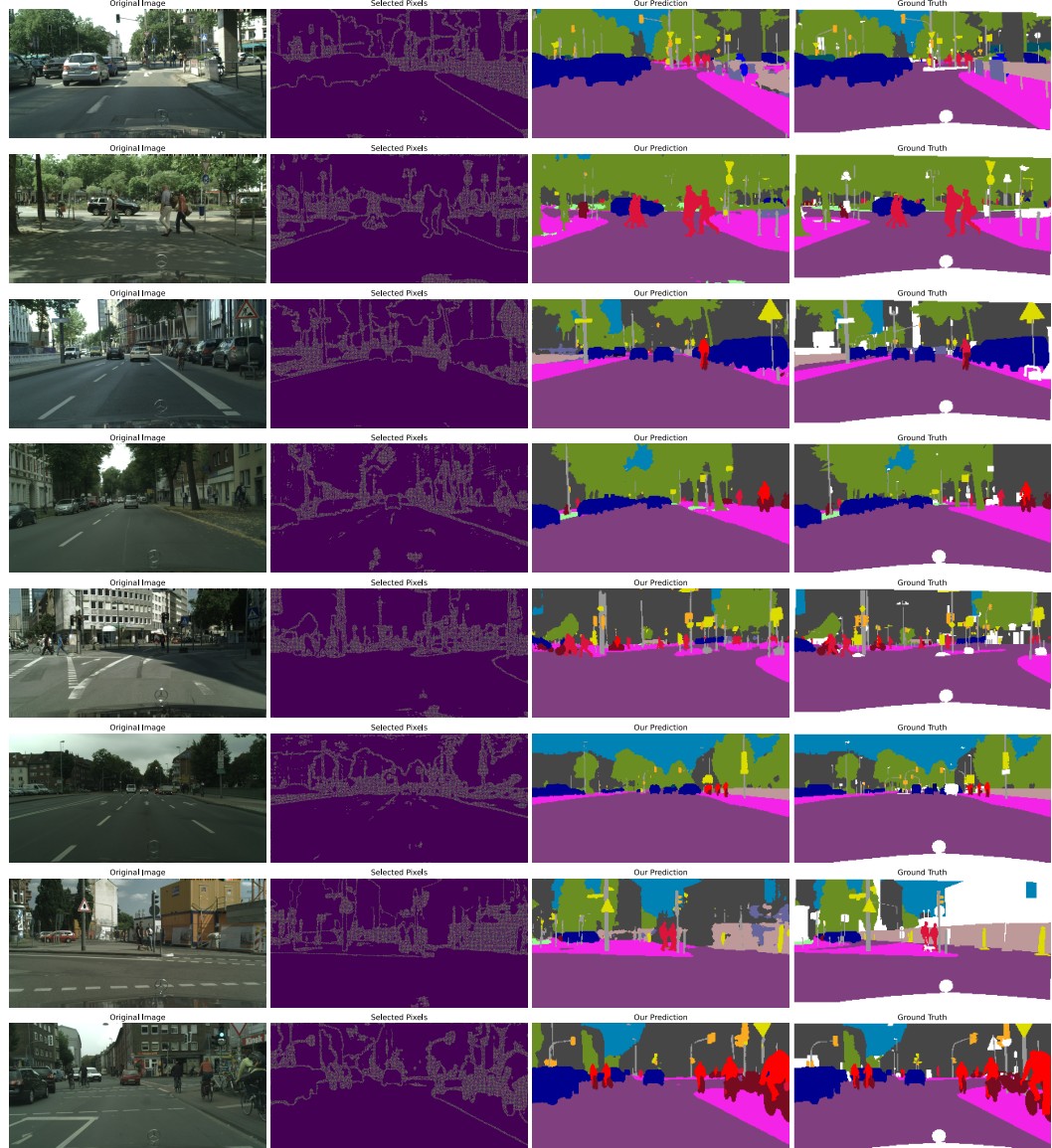

Figure 8: **Qualitative Results Visualization for the GTAV → Cityscapes Task.** The figure show-cases different subfigures representing: the original image, HALO's pixel selection, HALO's prediction, and the ground-truth label. Zoom in for the details.

## A.9 ADDITIONAL CORRELATION ANALYSES

**Correlation Analysis on Cityscapes → ACDC**    In addition to GTAV → Cityscapes, we report the correlations for Cityscapes → ACDC. The new correlations (hyperbolic radius Vs. class accuracy, hyperbolic radius Vs. percentage of target pixels) are (-0.759,-0.868). Compared to the case of GTAV → Cityscapes, (-0.605,-0.899), we note a steep rise in the correlation between the average per-class hyperbolic radius and the class accuracy. This is additional experimental evidence in favor of the novel geometric interpretation of the hyperbolic radius as an estimator of unexplained class complexity proposed in this paper.

**Class Imbalance with Increasing Budget**    We present an additional experiment tracking the evolution of class imbalance in the selected labels of the target dataset as the budget increases (refer to Fig. 10). When we start the acquisition, namely after having selected only 0.1% of labels from the

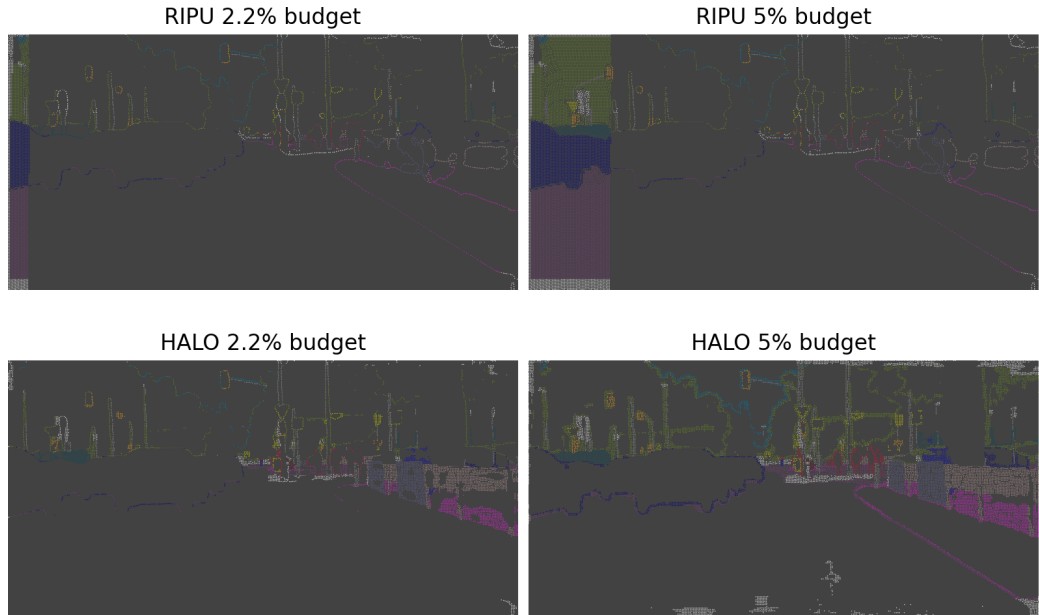

Figure 9: (top-row) Pixel selection with RIPU's baseline; (bottom-row) Pixel selection with out HALO; (left-column) Selection with budget 2.2%; (right-column) Selection with budget 5%. Zoom in for the details.

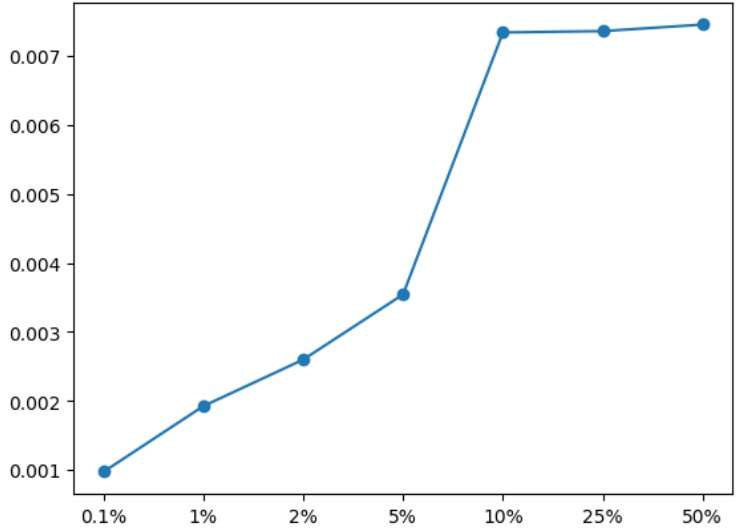

Figure 10: Evolution of the variance (y axis) of selected pixel distributions with varying budget (x axis).

target dataset, the variance is at a minimum, as HALO manages to identify and select labels from each class, in equal proportions. Then the variance increases slowly until the budget reaches 5%. This happens because the model manages to select pixels from each class, balancing the acquired data selection. The variance has a steep increase at budgets of 10% and higher. This occurs because the model has already selected most of the labels from the complex and scarce classes which it can identify thanks to the hyperbolic radius and the entropic uncertainty (cf. Sec. 5.2). So, for budgets of 10% or more, the model data acquisition strategy is influenced by the target dataset imbalance, which shows in the steep increase of the variance of the number of selected labels per class. The imbalance trend in label selection matches the performance variation in Fig. 5 of the paper. We

conclude therefore that HALO's label selection aids performance, beyond the fully-supervised selection, until the model manages to successfully identify complex and scarce classes, and until they are available in the target dataset.

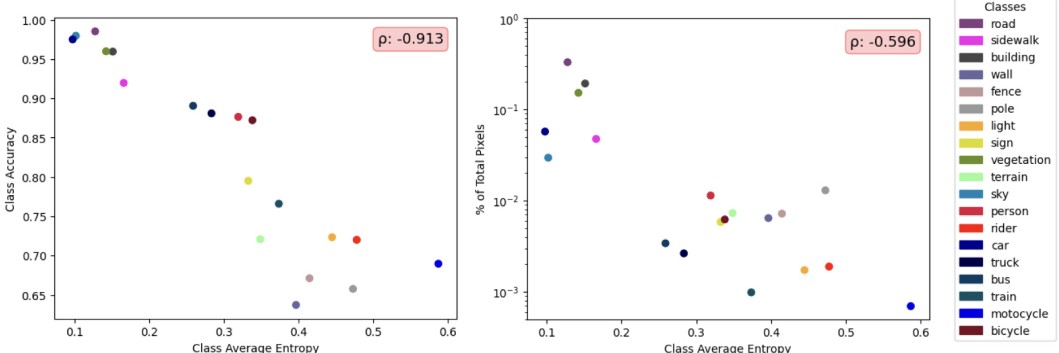

Figure 11: (left) Plot of average per-class entropy Vs. class accuracies; (right) Plot of average per-class entropy Vs. the percentage of per-class target pixels. Zoom in for the details.

**Correlation Analysis with the Entropy Score**    We report the correlations between the uncertainty (entropy of the classification probabilities) and the total number of pixels, and between the uncertainty and the accuracy. We consider uncertainty because HALO inherits it from RIPU (Xie et al., 2022a) and it complements it with the hyperbolic radius. In Fig. 11 we report the correlations between the class accuracy and the average class uncertainty (-0.913), and the correlation between the percentage of total pixels and the average class uncertainty (-0.596). The correlations of uncertainty Vs. accuracy and the percentage of total pixels are effectively large and complementary to those of the hyperbolic radius, which supports considering their combination, cf. Sec. 5.2.

