# OpenReview forum: "Hyperbolic Active Learning for Semantic Segmentation under Domain Shift"
_ICLR.cc/2024/Conference — Submitted to ICLR 2024_

### Official Review · Reviewer_3ouZ · 2023-10-29

**Soundness:** 2 fair
**Presentation:** 2 fair
**Contribution:** 2 fair
**Rating:** 5
**Confidence:** 4

**Summary:**

This paper propose to use the hyperbolic radius of each sample as selection criterion for active learning under domain shift. Specifically, they assume a hyperbolic image segmenter pretrained on the source domain. The segmenter is used to extract feature embedding for each pixel in the target domain, which is then projected to the hyperbolic space to compute the hyperbolic radius. The hyperbolic radius is then combined with entropy to select pixels for annotation. The proposed method is evaluated on several well-established domain adaptation settings for semantic segmentation, including GTAV→Cityscapes, SYNTHIA → Cityscapes, and Cityscapes → ACDC, and shown to outperform others.

**Strengths:**

1. The observation that the hyperbolic radius is correlated with class difficulty and scarcity is interesting.

**Weaknesses:**

1. The correlation between hyperbolic radius and class complexity is only supported with some experimental evidence on GTAV->Cityscapes dataset. Actually, as the two aspects of class complexity considered in the paper, i.e., class difficulty and scarcity, are correlated by itself, the hyperbolic radius may be mostly affected by label scarcity. The coefficient factor (-0.605 vs. -0.899) also indicates that the hyperbolic radius is more correlated with label scarcity. More convincing support (i.e., theoretical analysis, more experimental evidence) for the correlation between hyperbolic radius and class complexity is needed.
2. The writing needs to be improved, some sentences are broken, e.g., "So more difficult classes such as pole which have lower accuracy, also have the larger Riemannian variance, so the largest effective volume available." (available is not a verb), "Atigh et al. (2022) has been the first to demonstrate a performance of hyperbolic SS on par with Euclidean."(Euclidean is not a noun)

**Questions:**

1. How does the method perform on general AL tasks, e.g., AL for image classification on CIFAR10, CIFAR100, etc.

---

> ### Author Response · Authors · 2023-11-28
> **Response to Reviewer 3ouZ**
>
> Dear Reviewer 3ouZ,
> Your commitment to reviewing our paper is greatly appreciated. Kindly find our responses to your questions below:
>
> - **(Weakness 1)** This question plays a crucial role in understanding the novel interpretation of the hyperbolic radius and the proposed acquisition strategy, so we have addressed it in our **message to all reviewers**. Please see the answer on **Correlation Analysis**.
> Additionally, following the reviewer's advice, we have computed the correlation between the class accuracy and the percentage of total pixels, which yields 0.539. This correlation is moderate, but lower than that between the hyperbolic radius Vs.\ the class accuracy and total number of pixels, $-0.605$ and $-0.899$ respectively. In our answer on **Correlation Analysis**, as further support to the use of the hyperbolic radius as a data acquisition strategy, please also see the larger correlations in the case of Cityscapes $\rightarrow$ ACDC, -0.759 and -0.868 respectively.
>
> - **(Weakness 2)** Thank you for your comment. As highlighted in the **message to all reviewers**, we value your feedback and have diligently attended to the noted writing issues, conducting additional proofreading to enhance the overall quality of the paper. Check out the **Paper's writing and detailing** for the answer.
>
> - **(Question 1)** We agree on the intriguing potential of extending HALO for Active Learning to other tasks. In fact, we are actively engaged in research on active learning for image classification and object detection. However, we must acknowledge that this presents several challenges and complexities, which need careful considerations:
>   - **Divergent Task Requirements**: Our current work draws from the insights of Hyperbolic Image Segmentation (Atigh et al. 2022). For the object detection task an entirely distinct pipeline is necessary and no prior work is available. On the other hand, image classification with HNNs has proven effective primarily on datasets with inherent hierarchies and self-supervised learning approaches (Ermolov et al. 2022, HIER: Metric Learning Beyond Class Labels via Hierarchical Regularization Kim et al. 2022). Notably, the challenge arises from the fact that the significance of the hyperbolic radius was demonstrated at the pixel level for semantic segmentation tasks, whereas it becomes meaningful at the patch or full-image level primarily in the context of hierarchical relationships among classes. Our objective is to generalize the utility of hyperbolic learning without imposing hierarchical labels or relying on inherently hierarchical datasets with hyperbolic loss functions.
>   - **Dataset Imbalance**: An additional consideration lies in the observed high correlation between the hyperbolic radius and class difficulty in cases of highly imbalanced datasets. However, this correlation may not hold for datasets such as CIFAR-10/-100 or ImageNet, where class distributions tend to be more balanced.
>
>   Given these substantial disparities between the HNNs literature and the tasks at hand, extending HALO for active learning to these domains, while intriguing, would indeed represent a significant endeavor that surpasses the intended scope of this paper.

---

### Official Review · Reviewer_kZFp · 2023-11-01

**Soundness:** 3 good
**Presentation:** 2 fair
**Contribution:** 2 fair
**Rating:** 5
**Confidence:** 3

**Summary:**

The paper presents a novel approach called HALO (Hyperbolic Active Learning Optimization) for pixel-level active learning in semantic segmentation. The authors introduce a hyperbolic neural network method and a geometric interpretation of hyperbolic geometry derived from data statistics. In their approach, the hyperbolic radius represents an estimator of unexplained class complexity, which combines class intrinsic complexity and dataset scarcity. This complexity metric is used to identify the most informative pixels for annotation by considering both prediction uncertainty and class complexity. The approach is evaluated on several benchmarks and HALO achieves a new state-of-the-art in active learning for semantic segmentation, outperforming supervised domain adaptation with only 1% of labeled data.

**Strengths:**

- Hyperbolic NN for AL seems something new
- The method is interesting as the paper proposes to let the HALO learn a manifold where the distance of a class from the center is directly proportional to the unexplained class complexity.

**Weaknesses:**

- The paper is not that easy to follow as there are missing details like how to get the embeddings of the pixels? Directly in the pixel space or get the feature first? How to plot the Figure 2? How to get the accuracy in Figure 2?
- setting a new state-of-the-art across " all ADA benchmarks for SS" is overclaimed.
- This is not correct: "Hyperbolic neural networks first extract a feature vector v in Euclidean space". Not necessary in Euclidean space (Fully HNN).
- The conclusion seems problematic: We conclude that the hyperbolic radius indicates the difficulty in recognizing a class, as a consequence of the class complexity and its label scarcity.  Like in the previous papers, the hard pixels are more relative to their locations, like pixels on the boundary, even though there are lots of pixels in the dataset for such a category.

**Questions:**

- The paper mentioned that 'the pixels at the class boundaries are not necessarily the most informative and annotating only those degrade performance, as we confirm with an oracular study.' Then what should be the most informative ones? I ask because seems that the 'acquisition map' also indicates boundaries are the most informative ones?
- How can you ensure that the hyperbolic NN is measuring the scarcity of labels for certain class prototypical appearances and the intrinsic complexity of classes? As you know the uncertainty will also lead to such a state.
- How to make sure the manually defined class hierarchies are reliable? Why not learn from the data?
- It seems this is not true from Figure 2 left:  classes with larger hyperbolic radii have lower performance and are likely more difficult to recognize, and more complex. (BTW, there is no Fig (a) and (b), only left and right)

---

> ### Author Response · Authors · 2023-11-28
> **Response to Reviewer kZFp (part 1)**
>
> Dear Reviewer kZFp,
> Your dedicated time and effort in reviewing our paper are deeply appreciated. Enclosed are our responses addressing your questions and concerns:
>
> - **(Weakness 1)** We are sorry for the confusion caused by insufficient pipeline description and we appreciate the opportunity to clarify the details. Following [1,2], HALO comprises a Euclidean Segmenter (DeepLab-v2 or DeepLab-v3plus), a hyperbolic projection layer (expmap), and a hyperbolic multinomial logistic regression (HyperMLR) layer. The Euclidean Segmenter produces a _d_-dimensional embedding (d=64 for HALO) for each pixel of the image, which is projected onto the Poincaré ball via expmap, thus yielding the hyperbolic embedding of each pixel. The parameters to estimate the hyperbolic embedding for each pixel are end-to-end learned, based a pixel classification loss (cf. the HyperMLR layer.) To answer your question: embeddings are end-to-end learned in a hyperbolic manifold for each pixel. We have updated these details in the HALO pipeline in Sec. 5.1.
> Fig. 2 is generated by calculating the _Class Average Radius_, which involves computing the mean hyperbolic radius for all pixel embeddings within each class in the target dataset. Additionally, the _Class Accuracy_ in Fig. 2 (left) represents the average classification accuracy for each class in the target dataset. Finally, the _\% of Total Pixels_ in Fig. 2 (right) corresponds to the percentage of acquired pixel annotations at the end of the active learning process for each class.
>
> - **(Weakness 2)** To the best of our knowledge, RIPU [6] is the state-of-the-art on ADA for SS on the two benchmarks of GTAV $\rightarrow$ Cityscapes and SYNTHIA $\rightarrow$ Cityscapes. The two benchmarks were originally adopted by MADA [7] (Ning et al. ICCV'21), which proposes the task of ADA for SS for the first time. Our claim is supported by surpassing RIPU on both benchmarks and on a third one, which we additionally introduce, Cityscape $\rightarrow$ ACDC.
>
> - **(Weakness 3)** Let us note that Fully Hyperbolic Neural Networks (Fully-HNN) have not gained widespread adoption nor have they demonstrated to be on par with alternative approaches. We are aware of the following Fully-HNN: Poincaré ResNet [4], Hyperbolic RNN [5], and Hyperbolic MLP [5]. They do not yield the state-of-the-art in the benchmark tasks for which they were proposed. Also, none of them has been benchmarked on semantic segmentation, so far. This is likely due to issues on convergence and computational load, which is beyond the purpose of this paper to address. Our methodology is rooted in previous frameworks, such as [1,2,3], where the hyperbolic neural network is conventionally divided into an Euclidean model and subsequent hyperbolic projection and classification layers. In this process, a feature vector $v$ is indeed extracted from the Euclidean space before being projected into the Poincaré ball. This approach aligns with the current hyperbolic state-of-the-art method for semantic segmentation [1].

---

> ### Author Response · Authors · 2023-11-28
> **Response to Reviewer kZFp (part 2)**
>
> - **(Weakness 4 and Question 1)** Your inquiry is much appreciated as it regards the specific reason why HALO achieves state-of-the-art performance, in our view. Specifically, HALO's strategy involves the selection of boundary pixels for "easier" classes such as car or road, while also incorporating internal pixels for more complex classes such as person, rider, and bicycle. HALO showcases the importance of boundary pixels while acknowledging the significance of inner parts of objects in comprehending the scene. Please refer to Section A.6 in the appendix and Fig. 7 for more insights. To validate this, we conduct an oracle study involving the exclusive selection of boundary pixels during active learning stages. However, the resulting performance of 69.8\% mIoU was inferior to both HALO and RIPU. Please take a look at the "**Baseline selection comparison**" response in the **message to all reviewers** for more details about HALO's pixel selection.
>
> - **(Question 2)** Thank you for this question, as it allows us to remark the interesting insight of the proposed analysis. Usually, in hyperbolic space, when the (1 - radius) is interpreted as the uncertainty, a low radius indicates high uncertainty. However, in our case, classes with lower radii, such as road, sky, building, are actually easier to classify by the model (higher accuracy and mIoU), leading to a mismatch with the canonical interpretation of hyperbolic uncertainty. Effectively, our work had originated by the wish to leverage the relation between the hyperbolic radius and uncertainty, but it transformed into a novel geometric interpretation of the hyperbolic radius when we realized a novel interpretation was needed. The nuanced interpretation of the hyperbolic radius is extensively discussed in Sec. 4, where we also provide a comparison with existing viewpoints (see Sec. 4.2). While a theoretical proof remains still an open research question, our empirical evidence currently supports this novel interpretation.
>
> - **(Question 3)** Thank you for your question. Our approach is not dependent on manually defined class hierarchies. In fact, this study originates from retraining the architecture of [1] and finding that the interpretation of the hyperbolic radius changes when not enforcing hierarchies.
>
> 7. **(Question 4)** This query is crucial to understanding the empirical motivation of the novel interpretation of the hyperbolic radius and the proposed HALO data acquisition strategy, so we have addressed it in the **message to all reviewers**, under "**Correlation Analysis**". Please refer to it.
>
>
>
> [1] Atigh et al., "Hyperbolic Image Segmentation," CVPR 2022.
>
> [2] Ermolov et al., "Hyperbolic Vision Transformers: Combining Improvements in Metric Learning," CVPR 2022.
>
> [3] Franco et al., "Hyperbolic Self-Paced Learning for Self-Supervised Skeleton-based Action Representations," ICLR 2023.
>
> [4] van Spengler et al., "Poincaré ResNet," ICCV 2023.
>
> [5] Ganea et al., "Hyperbolic Neural Networks," NIPS 2018.
>
> [6] Xie et al., "Towards Fewer Annotations: Active Learning via Region Impurity and Prediction Uncertainty for Domain Adaptive Semantic Segmentation," CVPR 2022.
>
> [7] Ning et al., "Multi-Anchor Active Domain Adaptation for Semantic Segmentation", ICCV 2021

---

### Official Review · Reviewer_UCCT · 2023-11-20

**Soundness:** 3 good
**Presentation:** 2 fair
**Contribution:** 3 good
**Rating:** 6
**Confidence:** 3

**Summary:**

The authors introduced a Hyperbolic neural network to address the adaptive domain adaptation in the field of semantic segmentation. The proposed method HALO achieves SOTA across different benchmarks, which illustrates the benefits brought by HNN for ADA task on SS. The authors provided an interesting discussion towards Hyperbolic radius and the unexplained class complexity which may benefit the ADA community.

**Strengths:**

1.  It is interesting to see how Hypernolic neural network can benefit the active domain adaptation in semantic segmentation field.  The proposed method achieves SOTA performance compared with the leveraged baselines.

2. Comprehensive ablations are done with great insights towards the proposed method.

3. The motivation is well described. The proposed method is described clearly and easy to understand.

4. The authors provided an interesting discussion towards Hyperbolic radius and the unexplained class complexity which may benefit the ADA community.

**Weaknesses:**

1. The paper writing will limit this paper and still needs to be improved. For example, in Figure 3 and Figure 4, all the indexes ((a), (b), (c)...) are not marked on the Figures correspondingly. In text, the Figure is indicated by both Figure and Fig., which should be unified as the same. The authors are suggested to check the paper writing.

2. At the beginning of the Section 4.1, the authors claim that "Fig. 2a illustrates the correlation between the perclass average hyperbolic radius and the relative class SS accuracy.". However, the correlation between class accuracy and the class  average radius is not very obvious when acc < 0.8, the authors are suggested to add more analyses according to different accuracy ranges. More detailed analysis is suggested to be added. More analysis should be given for the Region- Vs. Pixel-based criteria part.

3. The authors are suggested to have a discussion regarding the number of the parameters of the proposed method with the leveraged baselines.

4. In Figure 8, the authors only provided the predictions of the proposed method,  the predictions from the baselines are also interesting to deliver more comparison with the proposed method.

5. In Section 5.3, the authors claim that "However, these approaches often yield suboptimal or comparable performances when compared to the Euclidean counterpart. ". Will it be possible to provide quantitative comparison between these approaches ((Guo et al., 2022; Franco et al., 2023; van Spengler et al., 2023)) and the HFR?

6. The method uses softmax score to serve as uncertainty. However, in model calibration field, pure softmax can barely estimate a good uncertainty score of the model without any open-set techniques, e.g., Monte-Carlo Dropout, OpenMax, Deep evidential learning,..., will the unsatisfied uncertainty prediction from the softmax score limit the performance of the proposed method? More discussion is expected toward this concern.

**Questions:**

Please refer to the weaknesses.

---

> ### Author Response · Authors · 2023-11-28
> **Response to Reviewer UCCT (part 1)**
>
> Dear Reviewer UCCT,
> We sincerely appreciate the time and effort you dedicated to reviewing our paper. Below, you'll find our responses to your questions and concerns:
>
> 1. Thanks for the feedback. We've addressed the writing issues and we've conducted thorough proofreading of the paper. This has involved rectifying the indexes of Figures 2, 3, and 4, ensuring consistent labeling with "Fig." We've also refined certain sentences to enhance the overall clarity of the paper.
>
>
> 2. We sincerely appreciate your insightful question and the opportunity to further elucidate the correlation analysis presented in Section 4.1. Please consider the answer "**Correlation Analysis**", reported in the **message to all reviewers**, where we present a more detailed analysis of the correlations.
> In this answer, we discuss some additional insights on the Region- Vs. Pixel-based criteria, following your suggestion. While the region impurity score of RIPU requires pixel regions, as the impurity is based on region statistics, the hyperbolic radius employed in HALO can be computed on both pixel and region bases. In response to your suggestion, we train HALO with the region-based approach. As we can observe in the following table, the region-based approach leads to a small difference of -0.4\% on the GTAV $\rightarrow$ Cityscapes benchmark with 5\% acquired labels, but still manages to achieve a significative improvement over the baseline (RIPU).
>
>     | Method | Pixel-based | Region-based |
>     |-------|-------|-------|
>     | RIPU | - | 71.2 |
>     | HALO | 74.5 | 74.1 |
>
> 3. Your query holds significant importance and offers valuable insight into comparing parameters between Euclidean and Hyperbolic Neural Networks. To delve deeper into this comparison, we conduct an assessment of the parameter count between our method HALO and the prior state-of-the-art, RIPU [Xie et al.]. Both employ the DeepLab-v3+ architecture but with some distinctions:
>     1. RIPU operates with a pixel embedding dimension of 512, resulting in a parameter count of 60.1M.
>     2. In contrast, HALO operates with a reduced pixel embedding dimension of 64, which the adoption of a hyperbolic learning enables. Moreover, the HyperMLR requires fewer parameters than the Euclidean Linear layer used for classification due to the reduced embedding dimension. This results in a slightly lower total parameter count than RIPU's (10k fewer params). Additionally, HALO introduces the HFR module, consisting of two linear layers separated by a BatchNorm layer and a ReLU. Thanks to the lower embedding dimensions, the input and output sizes of the HFR module are only 64-dimensional, adding less than 10k additional parameters. This roughly matches the number of parameters removed from the segmenter. These modifications result in the parameter count being nearly identical between the two methods (60.1M), aligning with other studies leveraging the DeepLab-v3+ architecture.
> Here's a table for a better visual comparison between the models:
>
>     | Method | Segmenter | Dim. | HFR (params) | Total Params |
>     |-------|-------|-------|-------|-------|
>     | RIPU | DeepLab-v3+ | 512 | Not used | 60.1M |
>     | HALO | Hyper-DeepLab-v3+ | 64 | 10k | 60.1M |

---

> ### Author Response · Authors · 2023-11-28
> **Response to Reviewer UCCT (part 2)**
>
> 4. We appreciate your comment, as it prompted us to enhance the comprehensiveness of our paper. In Appendix A.8, we have included additional comparisons with RIPU, that comprehend a qualitative assessment of the selection process and present quantitative results from the oracle experiment to elucidate the limitations of RIPU's boundary selection. Please refer to the "**Baseline Selection Comparison**" section in the **message to all reviewers** for the complete answer.
>
>
>
> 5. Thank you for raising this point, as it prompted us to conduct a more comprehensive evaluation of these approaches. In response, we test Guo et al., 2022's Feature Clipping method in our framework for comparison with our HFR. As shown in the table below, while Feature Clipping works and produces better results than the baseline RIPU, it still falls short of our HFR method (-1.2\%). Guo et al. utilize Feature Clipping to prevent vanishing gradients during backpropagation. Despite its simplicity, this technique restricts the model's representational capacity by clipping features, resulting in inferior performance compared to HFR. In the allotted rebuttal time, we have not tested the curriculum learning of [Franco et al., 2023] and the initialization approach of [van Spengler et al., 2023], because their adaptation to the ADA task is not straightforward.
> The curriculum learning in Franco et al. is specifically tailored for metric learning scenarios involving a hyperbolic loss, enabling training in hyperbolic space by utilizing cosine distance for improved initialization, gradually transitioning to the Poincaré loss. Our method does not involve comparing embeddings and leveraging the Poincaré loss. Similarly, the initialization approach in van Spengler et al. is designed explicitly for fully hyperbolic ResNets, particularly hyperbolic convolutions. As we do not employ hyperbolic convolutional layers, their initialization approach is not immediately suitable for our model.
>
>     | Method | mIoU |
>     |--------|-----------|
>     | **Hyperbolic Feature Reweighting (ours)** | **74.5** |
>     | Feature Clipping (Guo et al., 2022) | 73.3 |
>     | Initialization (van Spengler et al., 2023) | not compatible |
>     | Curriculum Learning (Franco et al., 2023) | not compatible |
>
>
> 6. We genuinely appreciate your question and the opportunity it provides to clarify certain aspects of our proposed method. In our paper, we utilize the entropy of the softmax probability array as a measure of uncertainty (ref. Eq. 7), drawing from the existing literature on ADA [Gal et al., 2017; Wang & Shang, 2014; Wang et al., 2016; Sinha et al., 2019; Xie et al., 2022b; Prabhu et al., 2021; Xie et al., 2022a], which currently yields state-of-the-art results. We acknowledge that relying solely on the pure softmax value may not accurately estimate model uncertainty, as you point out. However, please consider that our contribution regards the introduction of the hyperbolic radius to complement uncertainty as the data acquisition strategy. Maintaining the entropic softmax formulation of earlier literature and esp. of RIPU enables a fair comparison. We agree that uncertainty deserves more consideration in future research.

---

### Author Response · Authors · 2023-11-28
**Message to all reviewers (part 1)**

Esteemed reviewers, we extend our appreciation for the time and dedication you've committed to reviewing our paper. Your input has been significantly valuable in enhancing the quality of our work.

We are pleased to inform you that we've taken great care in addressing each of the questions and considerations outlined in your reviews. Here is a summary of the actions we have taken to address your shared concerns:

- **Correlation Analysis:** The raised point {has prompted us to conduct three new experiments, yielding new exciting insights and consolidating the paper, namely:

    1. In addition to GTAV $\rightarrow$ Cityscapes, we report the correlations for Cityscapes $\rightarrow$ ACDC. The new correlations (hyperbolic radius Vs. class accuracy, hyperbolic radius Vs. percentage of target pixels) are (-0.759,-0.868). Compared to the case of GTAV $\rightarrow$ Cityscapes, (-0.605,-0.899), we note a steep rise in the correlation between the average per-class hyperbolic radius and the class accuracy. This is additional experimental evidence in favor of the novel geometric interpretation of the hyperbolic radius as an estimator of unexplained class complexity proposed in this paper. We have added the results into appendix A.9 of the updated paper.
    2. We present an additional experiment tracking the evolution of class imbalance in the selected labels of the target dataset as the budget increases (refer to Fig. 10 in appendix A.9).  When we start the acquisition, namely after having selected only 0.1\% of labels from the target dataset, the variance is at a minimum, as HALO manages to identify and select labels from each class, in equal proportions. Then the variance increases slowly until the budget reaches 5\%. This happens because the model manages to select pixels from each class, balancing the acquired data selection. The variance has a steep increase at budgets of 10\% and higher. This occurs because the model has already selected most of the labels from the complex and scarce classes which it can identify thanks to the hyperbolic radius and the entropic uncertainty (cf. Sec. 5.2). So, for budgets of 10\% or more, the model data acquisition strategy is influenced by the target dataset imbalance, which shows in the steep increase of the variance of the number of selected labels per class. The imbalance trend in label selection matches the performance variation in Fig. 5 of the paper. We conclude therefore that HALO's label selection aids performance, beyond the fully-supervised selection, until the model manages to successfully identify complex and scarce classes, and until they are available in the target dataset.
    3. Motivated by the reviewers' observation that the correlation between the hyperbolic radius and the total number of pixels is larger than that between the hyperbolic radius and the accuracy, we compute the correlations between the uncertainty (entropy of the classification probabilities) and the total number of pixels, and between the uncertainty and the accuracy. We consider the uncertainty because HALO inherits it from RIPU [Xie et al.] and it complements it with the hyperbolic radius. In Fig. 11 of Appendix A.9  we report the correlations between the class accuracy and the average class uncertainty (-0.913), and the correlation between the percentage of total pixels and the average class uncertainty (-0.596). The correlations of uncertainty Vs. accuracy and the percentage of total pixels are effectively large and complementary to those of the hyperbolic radius, which supports considering their combination, cf. Sec. 5.2.

---

### Author Response · Authors · 2023-11-28
**Message to all reviewers (part 2)**

- **Baseline selection comparison:** Following the reviewers' suggestion for a more comprehensive comparison, we incorporate a qualitative selection comparison between HALO and the baseline approach, RIPU [Xie et al.], in the new Fig. 9, described in Appendix A.8 of the paper. The top row of Fig. 9 depicts label acquisition using the baseline RIPU method with budgets of 2.2\% (left) and 5\% (right). The bottom row illustrates visualizations with our proposed HALO using the same budgets. Noteworthy observations include:

    1. By design, RIPU only concentrates on selecting boundaries between semantic parts (ref. Fig. 9 top-left). However, since there are only a few (thin) boundary pixels, RIPU soon exhausts the pixel selection request. Next, when a larger budget is available, RIPU simply samples from the left side. The random selection still provides additional labels (ref. Fig. 9 top-right) and is a good baseline, cf. Table 3 of [Xie et al.], although not as good as HALO's acquisition strategy.
    2. By contrast, HALO showcases pixel selection from both boundaries and internal regions within semantic parts (ref. Fig. 9 bottom-left). Especially passing from 2.2\% to 5\% acquisition budget, HALO considers thick boundaries, so also parts of objects close to the boundaries, but also areas within objects, as it happens for wall, fence, pole, and sidewalk, cf. the right image part in the bottom-right of Fig. 9.

- **What happens if we select from ground truth boundaries only (oracle performance test):** We replace in RIPU the pseudo-labels with ground-truth ones, so we test an active learning acquisition strategy solely based on ground-truth boundary pixels. Although oracular, the experiment yields a performance drop of 1.4 mIoU (69.8 mIoU compared to RIPU's 71.2 mIoU), which validates HALO's label selection approach from non-boundary regions.

- **Paper's writing and detailing:** In response to your prompts, we have enhanced the paper's writing, we have refined the labeling of figures, and we have conducted additional proofreading, which is now reflected in the updated online PDF. Additionally, we have improved the pipeline in Sec. 5.1, better explaining the computation of the embeddings and the hyperbolic radius.

---

### Meta-Review · Area_Chair_axbE · 2023-12-05

**Metareview:**

Dear authors,

Thank you for submitting the draft. Majority of the reviewers have assigned rank 5: marginally below the acceptance threshold, and 1 reviewer has assigned rank 6 (confidence 3). After careful review of the comments and rebuttal, we agree that paper needs a revision before being accepted.
Suggestion:-
Main concerns are readability, contribution and proof of the claims regarding hyperbolic radius.
Main contribution of the paper comes from the claim that "hyperbolic radius emerges as an estimator of the unexplained class
complexity, which encompasses the class intrinsic complexity and its scarcity in the dataset."
In main paper support for this claim was quite restrictive, however, authors have expanded the experiments around this claim and added them to the appendix. These and the respective analysis should be part of the main paper.
One suggestion to the authors could be that since their claim about the hyperbolic radius is quite generic and not just constrained to the segmentation. Solidifying why its true will increase its application. Adding experiments around image classification could be helpful in this regard.  Please note above is just a suggestion.


regards

**Justification For Why Not Higher Score:**

Have not been able to convince reviewers.

**Justification For Why Not Lower Score:**

N/A

---

### Decision · Program_Chairs · 2024-01-16

Reject